# Spatial-Aware Reduction Framework:
# Towards Efficient and Faithful Visual State Space Models

**Jindi Lv** [* 1]  **Aoyu Li** [* 1]  **Yuhao Zhou** [1]  **Zheng Zhu** [2]  **Xiaofeng Wang** [2 3]  **Qing Ye** [1]
**Yueqi Duan** [3]  **Wentao Feng** [1]  **Jiancheng Lv** [1]

Project page: https://spatial-aware-reduction-framework.github.io/

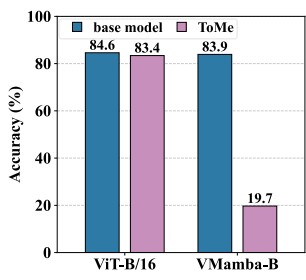
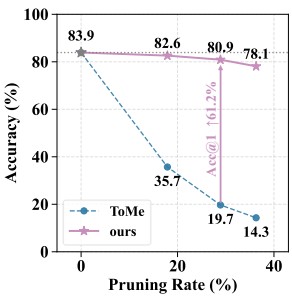
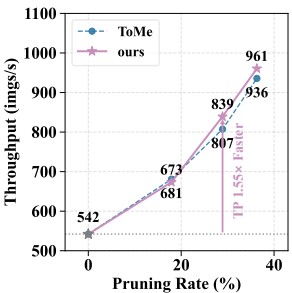
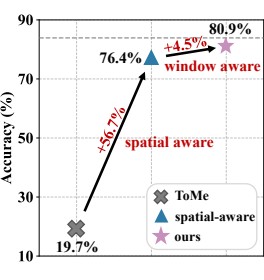

*(a)* Accuracy of token reduction: ViT-B vs. VMamba-B

*(b)* Accuracy comparison on VMamba-B: ToMe vs. ours.

*(c)* Throughput comparison on VMamba-B: ToMe vs. ours.

*(d)* Performance gains via spatial and window awareness

*Figure 1.* Motivation and efficacy of the proposed spatial-aware token reduction framework.

## Abstract

Mamba demonstrates strong efficiency in modeling long visual sequences. However, when token reduction is applied to structurally enhanced Mamba variants, these models exhibit a severe performance collapse. We attribute this degradation to the spatially agnostic nature of existing reduction methods, which violate the two-dimensional structural premise required by the selective scanning mechanism. In this work, we propose STORM, a spatial-aware token reduction framework designed to maintain structural integrity throughout the compression process. STORM reformulates reduction into a structured operation on spatial units, enforcing localized constraints to maintain both grid topology and neighborhood coherence. As a plug-and-play module, STORM equips existing reduction pipelines with explicit spatial awareness without any training. Empirical results demonstrate that STORM achieves state-of-the-art pruning accuracy across diverse vision Mamba backbones under training-free settings. Notably, STORM delivers a substantial accuracy recovery on VMamba, outperforming prior methods by up to 63.3% in top-1 accuracy. Meanwhile, STORM incurs only a 1.0% accuracy drop on PlainMamba, achieving performance comparable to ViT.

## 1. Introduction

Recently, Mamba architectures based on state space models (SSMs) (Gu et al., 2022; Smith et al., 2023; Gu & Dao, 2023; Fu et al., 2023) have emerged as an efficient solution for capturing long-range dependencies in vision tasks (Guo et al., 2024; Liang et al., 2024; Li et al., 2024; Zhu et al., 2024; Ma et al., 2024; Liu et al., 2024a; Ruan et al., 2024). By incorporating a selective scanning mechanism, Mamba effectively reduces computational complexity from the quadratic scaling of Transformers (Liu et al., 2021; Yin et al., 2022) to a linear regime. Building on this efficiency, integrating token reduction techniques such as pruning (Fayyaz et al., 2022; Rao et al., 2021; Yin et al., 2022; Kong et al., 2022; Katharopoulos et al., 2020) and merging (Bolya et al., 2023; Liang et al., 2022; Ryoo et al., 2021) offers a promising avenue to further optimize Mamba for practical deployment.

Nevertheless, existing token reduction methods exhibit limited compatibility with Mamba architectures compared to

[1]Sichuan University  [2]GigaAI  [3]Tsinghua University. Correspondence to: Wentao Feng <Wtfeng2021@scu.edu.cn>, Qing Ye <yeqing@scu.edu.cn>.

*Proceedings of the 43rd International Conference on Machine Learning*, Seoul, South Korea. PMLR 306, 2026. Copyright 2026 by the author(s).

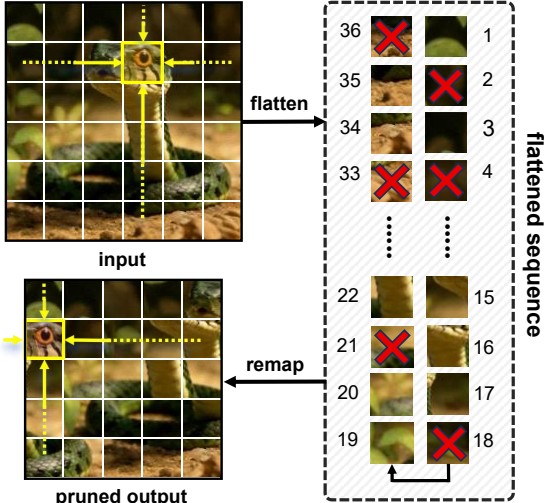

*Figure 2.* Pipeline of conventional token reduction in VMamba. The flattening operation explicitly disrupts spatial structure, causing misaligned representations after pruning.

Transformers ([Liu et al., 2021](); [Dosovitskiy et al., 2021](); [Touvron et al., 2021](); [Wang et al., 2021](); [Xu et al., 2022b](); [2023]()), as evidenced in Figure 1a. This degradation can be attributed to the fact that Transformers leverage self-attention to enable global token interactions, whereas Mamba relies on sequential scanning with strict recursive dependencies ([Lv et al., 2026]()). This chain-like structure renders Mamba highly susceptible to cascading information loss under token reduction, ultimately resulting in performance collapse.

In response, a new line of token reduction methods ([Zhan et al., 2024a]();[b](); [Ma et al., 2025](); [Shi et al., 2024a](); [Park et al., 2025]()) has been specifically developed for vision Mamba architectures. These methods perform well on vanilla Mamba architectures ([Zhu et al., 2024]()) but fail to generalize to the structurally enhanced variants ([Huang et al., 2024](); [Hatamizadeh & Kautz, 2025](); [Pei et al., 2025]()) such as VMamba ([Liu et al., 2024b]()). We attribute this limitation to an architectural evolution in VMamba, which introduces a 2D Selective Scan (SS2D) mechanism to strengthen global modeling. However, existing token reduction strategies remain spatially agnostic, violating the spatially sensitive structural assumptions required by SS2D.

To further elucidate this phenomenon, we present an illustrative example of token reduction on VMamba in Figure 2. In practice, these methods transform the 2D token layout into a sequential representation for reduction and then remap retained tokens back to a 2D structure to accommodate SS2D. This pipeline inevitably disrupts the original spatial relationships among tokens, resulting in spatial-semantic misalignment within the compressed representation. Inspired by this insight, we hypothesize that preserving the model's intrinsic structured representation during token reduction is critical

to maintaining architectural efficacy.

To validate this hypothesis, we introduce a structured reduction paradigm that operates on spatial primitives rather than globally flattened tokens. This approach preserves native structural integrity by independently reducing tokens within each row and column through two decoupled stages. As shown in Figure 1d, this simple paradigm shift yields a dramatic accuracy recovery from 19.7% to 76.4% under identical settings. This stark contrast confirms our insight that maintaining spatial consistency, not merely retaining salient tokens, is the key to effective compression in spatially-sensitive architectures.

Based on this principled shift, we propose **STORM**, a novel **S**patial-aware **TO**ken **R**eduction framework for vision **M**amba. By design, STORM serves as a seamless plug-in that equips existing reduction pipelines with structured spatial awareness. Building upon the above structured reduction paradigm, the framework further incorporates a localized windowing mechanism to protect fine-grained semantics. This mechanism confines reduction operations within coherent local neighborhoods, suppressing cross-region interference. By jointly enforcing global layout regularity and local contextual fidelity, STORM ensures representational integrity throughout the compression.

Empirical results validate STORM's effectiveness and generalizability in a training-free setting. As a versatile module, it enables substantial performance recovery on spatially-sensitive architectures and delivers consistent improvements across diverse Mamba backbones. As shown in Figures 1b and 1c, STORM consistently yields superior accuracy and throughput across all reduction ratios, notably surpassing ToMe ([Bolya et al., 2023]()) by **61.2%** in top-1 accuracy.

We highlight the main contributions of this paper below:

- We identify that spatial-structural integrity is the critical factor in performance degradation, and propose a structured reduction paradigm that operates on spatial primitives to effectively mitigate this issue.

- We propose a generic spatial-aware token reduction framework that incorporates localized neighborhood constraints to safeguard fine-grained semantics while preserving the benefits of organized spatial units.

- Our framework establishes a new state-of-the-art for vision Mamba reduction, consistently delivering superior accuracy and throughput across diverse backbones.

## 2. Related Work

### 2.1. Visual State Space Models

SSMs ([Gu & Dao, 2023](); [Mehta et al., 2023](); [Wang et al., 2023](); [Fu et al., 2023]()) were originally developed for se-

quence modeling in NLP to capture long-range dependencies with linear computational complexity, exemplified by S4's (Gu et al., 2022) structured diagonal state representations. To adapt SSMs for vision tasks, S4ND (Nguyen et al., 2022) introduced diagonal-normalized parameters but remained limited in capturing input-dependent spatial context. ViM (Zhu et al., 2024) addressed this by incorporating bidirectional scanning to enhance global awareness in visual representations. VMamba (Liu et al., 2024b) then introduced SS2D with four cross-directional scan paths, enabling comprehensive spatial modeling while maintaining linear computational efficiency. PlainMamba (Yang et al., 2024) further improved spatial modeling by adopting continuous 2D scanning that preserves token adjacency along the scanning path. More recent works have continued to refine spatial structure: Shi et al. (Shi et al., 2024b) proposed multi-scale 2D scanning by fusing features at multiple resolutions, and LocalMamba (Huang et al., 2024) adopted window-based scanning to better capture local spatial dependencies.

Despite these advances, vision Mamba models remain highly vulnerable to token reduction. In particular, spatially agnostic pruning inevitably disrupts the token connectivity required for their chain-like state propagation, leading to severe information loss. Although methods such as AMVim (Lv et al., 2026) improve robustness through multi-scale asymmetric scanning, this strategy cannot effectively generalize to the spatially structured architectural paradigm of vision Mamba. Consequently, the absence of a robust token reduction formulation remains a major bottleneck for the practical deployment of vision Mamba models.

### 2.2. Token Reduction

Token reduction improves computational efficiency by removing or consolidating redundant tokens during inference. Most existing approaches are developed for ViTs (Touvron et al., 2021; Wang et al., 2021; Bao et al., 2022) and can be broadly divided into token pruning (Rao et al., 2021; Yin et al., 2022; Kong et al., 2022; Liang et al., 2022) and token merging (Bolya et al., 2023; Chen et al., 2023; Marin et al., 2021; Xu et al., 2022a; Ryoo et al., 2021). Pruning methods discard low-importance tokens, as in DynamicViT (Rao et al., 2021) with a Gumbel Softmax strategy or EViT (Liang et al., 2022) using classification token attention. Merging methods reduce token counts by aggregating semantically similar tokens, exemplified by the training-free ToMe (Bolya et al., 2023) approach based on bipartite matching. Despite their success in ViTs, these techniques cannot be directly transferred to Mamba due to fundamentally different underlying mechanisms.

Recent work has explored token reduction specifically for vision Mamba architectures. For example, Zhan et al. (Zhan

et al., 2024a) proposed a pruning-aware hidden state alignment strategy to selectively skip less important tokens. In subsequent work, they introduced a hybrid criterion that jointly models token importance and similarity (Zhan et al., 2024b). More recently, MTR (Ma et al., 2025) was designed to estimate token importance by classifying tokens into three predefined categories. Although these methods show promise on vanilla Mamba models (Zhu et al., 2024), they remain limited in their ability to preserve the structural spatial relationships essential for more advanced variants (Liu et al., 2024b; Yang et al., 2024; Huang et al., 2024; Xie et al., 2024). To this end, instead of designing yet another specialized reduction method, we propose a spatial-aware framework that functions as a seamless wrapper for existing approaches. By equipping established reduction criteria with spatial intelligence, our framework maintains robust performance even under aggressive pruning ratios.

## 3. Method

### 3.1. Preliminaries

Mamba architectures are built upon discretized SSMs, which provide an efficient recurrent formulation for sequence modeling. A continuous-time SSM, parameterized by $(\mathbf{A}, \mathbf{B}, \mathbf{C})$, is converted into a discrete recurrence via zero-order hold discretization with step size $\Delta$:

$$
\begin{aligned}
h_t &= \overline{\mathbf{A}} h_{t-1} + \overline{\mathbf{B}} x_t, \\
y_t &= \mathbf{C} h_t,
\end{aligned}
\tag{1}
$$

where $\overline{\mathbf{A}} = \exp(\Delta \mathbf{A})$ and $\overline{\mathbf{B}} = (\Delta \mathbf{A})^{-1}(\exp(\Delta \mathbf{A}) - \mathbf{I}) \cdot \Delta \mathbf{B}$. Building on this formulation, Mamba introduces selectivity by conditioning $(\mathbf{B}, \mathbf{C}, \Delta)$ on the input $x_t$, enabling adaptive and data-dependent state transitions with linear computational complexity.

To adapt this sequential model to 2D visual data, architectures such as VMamba employ a SS2D mechanism. Given a 2D feature map $\mathbf{X} \in \mathbb{R}^{H \times W \times C}$, SS2D performs ordered scans along complementary spatial dimensions to construct a structured state transition graph. A horizontal scan processes each row $i$ from left to right:

$$
h_{i,j}^{\rightarrow} = \overline{\mathbf{A}} \odot h_{i,j-1}^{\rightarrow} + \overline{\mathbf{B}} \odot \mathbf{X}_{i,j}, \quad j = 1, \ldots, W, \tag{2}
$$

where $\odot$ denotes element-wise multiplication (parametrizing selectivity) and $h_{i,0}^{\rightarrow} = \mathbf{0}$. A subsequent vertical scan processes each column $j$ from top to bottom:

$$
h_{i,j}^{\downarrow} = \overline{\mathbf{A}} \odot h_{i-1,j}^{\downarrow} + \overline{\mathbf{B}} \odot \mathbf{X}_{i,j}, \quad i = 1, \ldots, H, \tag{3}
$$

with $h_{0,j}^{\downarrow} = \mathbf{0}$. The outputs from both directions are fused to form the final representation:

$$
\mathbf{Y}_{i,j} = \mathbf{C} \odot h_{i,j}^{\rightarrow} + \mathbf{C} \odot h_{i,j}^{\downarrow}. \tag{4}
$$

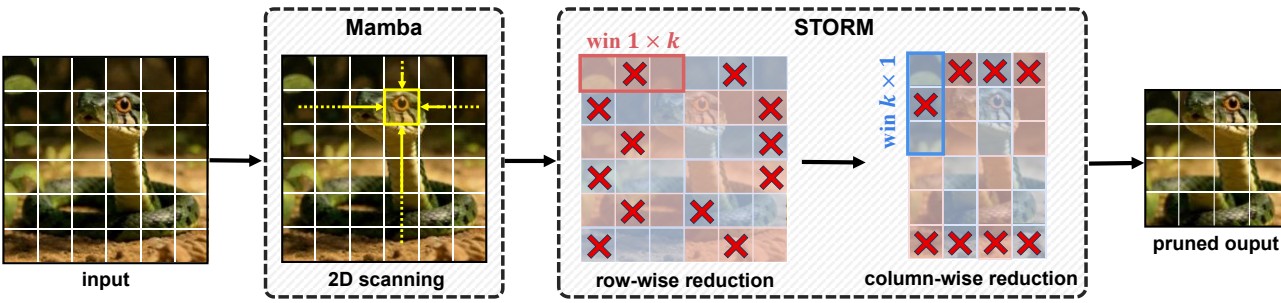

*Figure 3.* Overview of STORM. The framework performs spatially structured token reduction in two decoupled stages: row-wise and then column-wise reduction within localized windows, preserving the 2D grid layout required for selective scanning.

This process tightly aligns state evolution with the spatial logic of the 2D lattice, establishing an inherently spatial-aware computational pattern. This design imposes a fundamental premise: the model's efficacy necessitates an intact and regular token grid topology. Disrupting this spatial structure irrecoverably compromises the causal state propagation defined by the scanning order.

### 3.2. Overall Framework

In this work, we propose STORM, a lightweight framework that refactors token reduction into a spatially structured process, as illustrated in Figure 3. By design, STORM explicitly preserves the grid structure assumed by SS2D-based reasoning, ensuring alignment with the underlying computation pattern.

The framework enforces a dimensionally decomposed workflow that processes tokens sequentially along rows and columns. This design mirrors the directional scanning logic of SS2D, ensuring that the compressed tokens form a regular sub-grid whose spatial adjacency remains aligned with the model's intrinsic state propagation paths. To protect local semantics and suppress cross-region interference, STORM incorporates a localized windowing mechanism that confines reduction decisions to non-overlapping segments along each spatial dimension. This preserves neighborhood coherence while supporting effective compression. Importantly, within each stage, all row or column operations are executed in parallel, maintaining high efficiency.

By integrating structured spatial reduction with localized constraints, STORM enables existing token reduction methods to operate in harmony with the spatial premises of vision Mamba models, achieving both computational efficiency and robust representational integrity.

### 3.3. Structured Spatial Reduction

The structured spatial reduction component implements the core operational shift from globally flattened token processing to dimension-wise reduction within the feature grid.

Given an input feature map $\mathbf{X} \in \mathbb{R}^{H \times W \times C}$, the goal is to produce a compressed output $\mathbf{X}' \in \mathbb{R}^{H' \times W' \times C}$ that preserves the exact topological relationships required for faithful recurrent scanning.

We define a generic reduction operator $\mathcal{R}(\cdot)$ that encapsulates any base token reduction method, such as importance-based pruning or similarity-driven merging. Reduction is performed in two successive stages aligned with the spatial dimensions of the grid, first along rows and then along columns, ensuring the output remains a regular sub-grid.

**Horizontal (Row-wise) reduction.** In the first stage, the feature map is decomposed into row-wise sequences by fixing the row index $i$. Each row is represented as

$$\mathbf{X}_{i,:} = \{\mathbf{X}_{i,j}\}_{j=1}^{W}, \quad \mathbf{X}_{i,:} \in \mathbb{R}^{W \times C}.$$

The reduction operator $\mathcal{R}$ is applied independently to each row,

$$\widetilde{\mathbf{X}}_{i,:} = \mathcal{R}(\mathbf{X}_{i,:}), \quad \widetilde{\mathbf{X}}_{i,:} \in \mathbb{R}^{W' \times C}. \tag{5}$$

Stacking all reduced rows along the height dimension yields an intermediate feature map $\widetilde{\mathbf{X}} \in \mathbb{R}^{H \times W' \times C}$.

**Vertical (Column-wise) reduction.** In the second stage, the intermediate feature map $\widetilde{\mathbf{X}}$ is further decomposed into column-wise sequences by fixing the column index $j$. Each column is given by

$$\widetilde{\mathbf{X}}_{:,j} = \{\widetilde{\mathbf{X}}_{i,j}\}_{i=1}^{H}, \quad \widetilde{\mathbf{X}}_{:,j} \in \mathbb{R}^{H \times C}.$$

The same reduction operator $\mathcal{R}$ is applied independently to each column,

$$\mathbf{X}'_{:,j} = \mathcal{R}(\widetilde{\mathbf{X}}_{:,j}), \quad \mathbf{X}'_{:,j} \in \mathbb{R}^{H' \times C}. \tag{6}$$

Aggregating all reduced columns produces the final output $\mathbf{X}' \in \mathbb{R}^{H' \times W' \times C}$.

Through this formulation, the output forms a contiguous sub-grid whose topology aligns perfectly with the scanning paths assumed by the underlying SS2D mechanism. Consequently, the structured reduction preserves the causal dependencies of state propagation, enabling compression without disrupting the model's learned dynamics.

## 3.4. Localized Window Mechanism

While structured spatial reduction preserves the global grid topology required by SS2D, unconstrained reduction within an entire row or column may still introduce long-range interference that disrupts local semantic coherence. In SS2D, state propagation is inherently sequential and locality-sensitive, as each state update depends on a compact neighborhood along the scanning path. Reduction decisions that span distant spatial regions therefore risk collapsing semantically unrelated tokens, leading to distorted state transitions.

To address this issue, STORM introduces a localized window mechanism that constrains reduction to spatially contiguous neighborhoods. For row-wise reduction, each row sequence $\mathbf{X}_{i,:} \in \mathbb{R}^{W \times C}$ is partitioned into $K$ non-overlapping windows,

$$\mathbf{X}_{i,:} = \bigcup_{k=1}^{K} \mathbf{X}_{i,:}^{(k)}, \quad \mathbf{X}_{i,:}^{(k)} \in \mathbb{R}^{L \times C}, \tag{7}$$

where $L$ denotes the window length and $K = W/L$. Reduction is applied independently within each window,

$$\widetilde{\mathbf{X}}_{i,:}^{(k)} = \mathcal{R}\big(\mathbf{X}_{i,:}^{(k)}\big), \quad \widetilde{\mathbf{X}}_{i,:}^{(k)} \in \mathbb{R}^{L' \times C}, \tag{8}$$

and the reduced windows are concatenated in their original order to form the reduced row

$$\widetilde{\mathbf{X}}_{i,:} = \bigcup_{k=1}^{K} \widetilde{\mathbf{X}}_{i,:}^{(k)}, \quad \widetilde{\mathbf{X}}_{i,:} \in \mathbb{R}^{W' \times C}. \tag{9}$$

An analogous procedure is applied during column-wise reduction. Each column sequence $\widetilde{\mathbf{X}}_{:,j} \in \mathbb{R}^{H \times C}$ is partitioned into non-overlapping vertical windows, and the same reduction operator $\mathcal{R}$ is applied independently within each window. The reduced windows are concatenated in their original order to form $\mathbf{X}'_{:,j} \in \mathbb{R}^{H' \times C}$, ensuring that reduction decisions remain confined to spatially coherent neighborhoods along the vertical scanning dimension.

By confining reduction to localized windows, STORM preserves the short-range spatial dependencies essential for stable state evolution in SS2D. This mechanism safeguards local contextual coherence, complementing the global topological regularity enforced by structured spatial reduction. Together, dimension-wise processing and localized windowing enable STORM to perform aggressive token reduction while faithfully respecting the structural and semantic foundations of SS2D-based vision Mamba architectures.

## 3.5. Theoretical Analysis

To investigate why global reduction collapses performance and how STORM guarantees stability, we analyze VSSM hidden state dynamics under token compression.

**Error Formulation.** Following Sec. 3.1, the state update is $h_t = \overline{\mathbf{A}}_t h_{t-1} + \overline{\mathbf{B}}_t x_t$. Let $\mathcal{S} = \{x_1, \ldots, x_T\}$ be the sequence from grid $\mathbf{X}$. Token reduction yields a compressed sequence $\hat{\mathcal{S}} = \{\hat{x}_1, \ldots, \hat{x}_{T'}\}$ $(T' < T)$ and a corrupted state $\hat{h}_t = \overline{\mathbf{A}}_t \hat{h}_{t-1} + \overline{\mathbf{B}}_t \hat{x}_t$. Assuming a stable parameterization where $\| \prod_{k=\tau+1}^{t} \overline{\mathbf{A}}_k \|_2 \leq \gamma^{t-\tau}$ ($\gamma < 1$), the final state error $\mathcal{E}_T = \mathbb{E}[\|\hat{h}_T - h_T\|^2]$ is upper-bounded by:

$$\mathcal{E}_T \leq C \sum_{t=1}^{T'} \gamma^{T'-t} \cdot \mathbb{E}[\|\hat{x}_t - x_{\mathcal{M}(t)}\|^2], \tag{10}$$

where $\mathcal{M}(t)$ maps the reduced index to its original position, and $C = \max_t \|\overline{\mathbf{B}}_t\|_2^2$.

**Dilemma of Agnostic Methods.** We define the spatial displacement as $\mathcal{D}(x_i, x_j) = \|p(x_i) - p(x_j)\|_2$, where $p(\cdot)$ outputs 2D lattice coordinates. Standard 2D images inherently possess spatial locality, where tokens within a local neighborhood share coherent semantic features.

Global methods (e.g., ToMe, EViT) cluster tokens based solely on feature similarity. Because they lack spatial constraints, they routinely merge non-local tokens and align them sequentially after reduction. This forces the scanning trajectory to perform long-range spatial jumps, rendering the displacement between consecutive tokens $\mathcal{D}(\hat{x}_{t-1}, \hat{x}_t)$ unconstrained. By disrupting the 2D grid topology, the local representation continuity is completely shattered. Consequently, the individual token reconstruction error $\mathbb{E}[\|\hat{x}_t - x_{\mathcal{M}(t)}\|^2]$ scales unpredictably with the severity of trajectory scrambling, causing the cumulative error bound in Eq. (9) to diverge and leading to catastrophic performance collapse.

**Error Bounding via STORM.** STORM resolves this vulnerability via two geometric constraints: 1) *Dimension Decomposition:* Separating reduction into row and column sub-grids (Sec. 3.3) ensures $\hat{\mathcal{S}}$ remains a structurally valid sub-lattice, preventing arbitrary trajectory permutations. 2) *Window Constraints:* Although reduction metrics are computed globally, compression actions are restricted within local windows $\Omega_k$ of length $L$ (Sec. 3.4). The maximal mapping displacement is strictly bounded by the window scale:

$$\max_{t \in \Omega_k} \mathcal{D}(\hat{x}_t, x_{\mathcal{M}(t)}) \leq \alpha \cdot L, \tag{11}$$

where $\alpha$ is the pixel pitch. This strictly preserves spatial locality. Since the maximum reconstruction discrepancy is bounded by a monotonic function of the window size, $\max_t \mathbb{E}[\|\hat{x}_t - x_{\mathcal{M}(t)}\|^2] \leq g(\alpha L)$, substituting this into Eq.(9) establishes a tight upper bound:

$$\mathcal{E}_T \leq C \cdot \frac{1 - \gamma^{T'}}{1 - \gamma} \cdot g(\alpha L). \tag{12}$$

As $L$ remains small, $g(\alpha L) \to 0$, rigorously suppressing error accumulation and guaranteeing stable state propagation.

# 4. Experiments

## 4.1. Datasets and Settings

We conduct comprehensive experiments on the ImageNet-1K (Deng et al., 2009) and COCO 2017 (Lin et al., 2014) datasets. To examine the generalization of STORM, we evaluate it on multiple Mamba backbones, including VMamba (Liu et al., 2024b), PlainMamba (Yang et al., 2024), and LocalMamba (Huang et al., 2024). The effectiveness of STORM is evaluated on both token pruning and token merging methods, specifically EViT (Liang et al., 2022) and ToMe (Bolya et al., 2023).

All reported results are obtained under a training-free setting. Unless otherwise specified, the default window size of STORM is set to 5. All experiments are conducted using 4 × NVIDIA RTX 4090 GPUs. Implement details are provided in the Appendix 6.

## 4.2. Image Classification

Table 1 compares the performance of token reduction methods on ImageNet-1K classification under training-free settings. The results show that STORM consistently achieves the highest accuracy across all model scales and backbones, demonstrating superior robustness to compression.

STORM significantly mitigates the performance drop caused by token reduction. On VMamba-S, STORM (ToMe) limits accuracy loss to 2.8%, compared to a 50.8% drop with standard ToMe. On PlainMamba, STORM reduces the gap to just 1.0%. This improvement stems from STORM's structured reduction paradigm, which preserves the 2D spatial layout required by the scanning mechanism, maintaining model integrity where spatially-agnostic methods fail. Results on LocalMamba are provided in Appendix 7.1.

## 4.3. Object Detection and Instance Segmentation

Table 2 compares the performance of various token reduction methods on object detection and instance segmentation tasks. The results show that our STORM framework consis-

tently achieves the best performance across both tasks on all evaluated Mamba backbones, substantially outperforming prior methods under training-free settings.

STORM enables a dramatic accuracy recovery on VMamba, elevating its detection AP$^b$ score from a collapsed state near 3.7 to a competitive 46.2. On PlainMamba, STORM also provides a clear and consistent improvement over baseline reduction methods. These results demonstrate that STORM delivers superior robustness in downstream dense prediction tasks, effectively preserving model performance during compression. Tiny-scale results are reported in Appendix 7.2.

## 4.4. Robustness Analysis

**Comparison with SOTA methods.** Table 3 compares the performance of recent Mamba-specific token reduction methods on ImageNet-1K classification under training-free settings. The results demonstrate that STORM framework achieves the highest top-1 accuracy across both model scales while maintaining comparable computational efficiency.

STORM recovers nearly all of the baseline accuracy, reaching 82.0% and 83.3% on the small and base models, respectively. In stark contrast, even the latest dedicated methods suffer severe degradation: HSA (Zhan et al., 2024a) drops to 46.2%, MTR (Ma et al., 2025) reaches only 47.9%, and QuarterMap (Chi et al., 2025) achieves only 64.2% on the small model. This decisive advantage stems from STORM's core design. While other methods are tailored to the scan mechanism, they remain spatially agnostic. STORM explicitly preserves the critical 2D token grid, aligning compression with the model's structural bias and delivering substantially more robust performance.

**Robustness across various reduction ratios.** Figure 4 (a) compares the accuracy of ToMe and STORM (ToMe) on VMamba-S across increasing reduction ratios. STORM maintains significantly higher accuracy under aggressive compression, overcoming the inherent matching limit of ToMe's global bipartite algorithm. Notably, STORM retains 71.7% accuracy at a 40% reduction rate, surpassing the

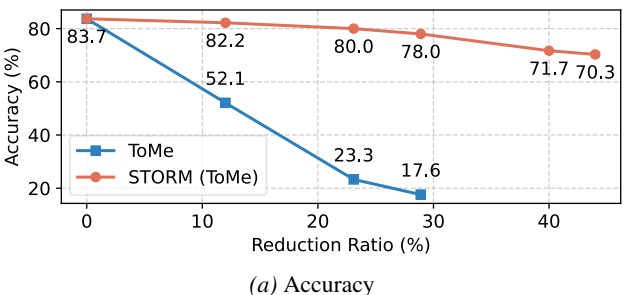

*(a) Accuracy*

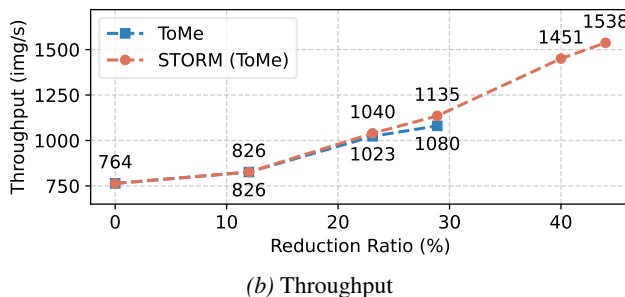

*(b) Throughput*

*Figure 4.* Comparison with conventional token reduction methods on VMamba-S in terms of top-1 accuracy and throughput across varying reduction ratios. STORM shows markedly stronger robustness than ToMe while enabling more aggressive reduction.

*Table 1.* Performance comparison on ImageNet-1K. The symbol "+" denotes the application of token reduction methods to off-the-shelf models, with training-free accuracy reported. The symbol △ indicates the performance difference between the baseline and the reduced model. STORM consistently yields substantial improvements over traditional reduction methods across diverse models.

| Method | GFlops | Params (M) | Acc1 (%) | △ (%) |
|---|---|---|---|---|
| VMamba | | | | |
| VMamba-T | 4.91 | 30 | 82.6 | 0.0 |
| +EViT | 3.00 | 30 | 15.2 | 67.4↓ |
| +ToMe | 3.27 | 30 | 31.5 | 51.1↓ |
| +STORM (EViT) | 3.00 | 30 | **78.5** | **4.1**↓ |
| +STORM (ToMe) | 3.00 | 30 | **78.8** | **3.8**↓ |
| VMamba-S | 8.72 | 50 | 83.7 | 0.0 |
| +EViT | 5.30 | 50 | 23.0 | 60.7↓ |
| +ToMe | 5.57 | 50 | 32.9 | 50.8↓ |
| +STORM (EViT) | 5.31 | 50 | **80.5** | **3.2**↓ |
| +STORM (ToMe) | 5.31 | 50 | **80.9** | **2.8**↓ |
| VMamba-B | 15.36 | 89 | 83.9 | 0.0 |
| +EViT | 9.33 | 89 | 24.4 | 59.5↓ |
| +ToMe | 9.69 | 89 | 35.7 | 48.2↓ |
| +STORM (EViT) | 9.33 | 89 | **82.2** | **1.7**↓ |
| +STORM (ToMe) | 9.33 | 89 | **82.6** | **1.3**↓ |
| PlainMamba | | | | |
| PlainMamba-L1 | 3.02 | 7 | 77.7 | 0.0 |
| +EViT | 2.45 | 7 | 66.1 | 11.6↓ |
| +ToMe | 2.46 | 7 | 69.6 | 8.1↓ |
| +STORM (EViT) | 2.45 | 7 | **75.3** | **2.4**↓ |
| +STORM (ToMe) | 2.45 | 7 | **75.4** | **2.3**↓ |
| PlainMamba-L2 | 8.08 | 26 | 81.6 | 0.0 |
| +EViT | 6.42 | 26 | 76.6 | 5.0↓ |
| +ToMe | 6.43 | 26 | 76.7 | 4.9↓ |
| +STORM (EViT) | 6.42 | 26 | **80.5** | **1.1**↓ |
| +STORM (ToMe) | 6.42 | 26 | **80.6** | **1.0**↓ |
| PlainMamba-L3 | 14.44 | 51 | 82.2 | 0.0 |
| +EViT | 9.74 | 51 | 75.2 | 7.0↓ |
| +ToMe | 9.75 | 51 | 76.1 | 6.1↓ |
| +STORM (EViT) | 9.74 | 51 | **80.1** | **2.1**↓ |
| +STORM (ToMe) | 9.74 | 51 | **80.9** | **1.3**↓ |

*Table 2.* Performance comparison on object detection and instance segmentation. The symbol "+" denotes the application of token reduction methods to off-the-shelf models under training-free settings. STORM achieves the best performance across both tasks, with particularly strong results on the VMamba backbone.

| Method | $AP^b$ | $AP^b_{50}$ | $AP^b_{75}$ | $AP^m$ | $AP^m_{50}$ | $AP^m_{75}$ |
|---|---|---|---|---|---|---|
| VMamba | | | | | | |
| VMamba-S | 48.7 | 70.0 | 53.5 | 43.7 | 67.3 | 47.0 |
| +EViT | 3.6 | 10.0 | 2.0 | 1.3 | 4.0 | 0.6 |
| +ToMe | 2.8 | 7.8 | 1.5 | 0.9 | 2.8 | 0.5 |
| +STORM (EViT) | **44.7** | **69.7** | **50.6** | **40.4** | **66.6** | **43.0** |
| +STORM (ToMe) | **44.2** | **69.5** | **49.8** | **39.8** | **66.0** | **42.1** |
| VMamba-B | 49.2 | 71.4 | 54.0 | 44.1 | 68.3 | 47.7 |
| +EViT | 3.7 | 10.5 | 2.0 | 1.3 | 4.2 | 0.5 |
| +ToMe | 2.8 | 8.2 | 1.4 | 0.9 | 3.1 | 0.4 |
| +STORM (EViT) | **46.2** | **70.7** | **52.0** | **40.9** | **67.5** | **43.9** |
| +STORM (ToMe) | **45.7** | **70.6** | **51.3** | **40.4** | **67.0** | **43.3** |
| PlainMamba | | | | | | |
| PlainMamba-L2 | 46.0 | 66.9 | 50.1 | 40.6 | 63.8 | 43.6 |
| +EViT | 34.9 | 54.3 | 37.3 | 31.1 | 51.3 | 32.7 |
| +ToMe | 33.7 | 53.1 | 35.6 | 30.2 | 50.2 | 31.6 |
| +STORM (EViT) | **39.1** | **61.8** | **42.0** | **34.7** | **57.9** | **36.4** |
| +STORM (ToMe) | **40.7** | **63.0** | **43.5** | **35.7** | **59.1** | **37.4** |
| PlainMamba-L3 | 46.8 | 68.0 | 51.1 | 41.2 | 64.7 | 43.9 |
| +EViT | 36.4 | 55.3 | 39.3 | 32.6 | 52.5 | 34.7 |
| +ToMe | 35.4 | 54.2 | 38.0 | 31.5 | 51.5 | 33.0 |
| +STORM (EViT) | **39.4** | **62.2** | **41.9** | **34.8** | **58.3** | **36.6** |
| +STORM (ToMe) | **40.8** | **63.8** | **43.7** | **35.9** | **59.7** | **37.9** |

**Robustness on random reduction.** Table 4 compares the performance of different token reduction strategies under training-free settings. The results demonstrate that STORM significantly outperforms the conventional ToMe method across all reduction ratios. Notably, even when employing a random token selection strategy, STORM still maintains robust performance. At a high reduction ratio of 36%, STORM (random) preserves 69.1% accuracy, a stark contrast to the 11.7% achieved by ToMe.

This resilience stems from STORM's structured design. The method preserves the 2D token layout during compression, ensuring the model's scanning mechanism remains functional. Thus, performance depends more on spatial integrity than on the specific token selection criterion, making STORM a fundamentally robust reduction framework.

**Comparison with conventional spatial reduction.**

Figure 5 compares STORM with conventional spatial reduction methods, pooling and downsampling, across varying token reduction ratios. The results show that STORM maintains significantly higher accuracy under aggressive compression, outperforming both pooling and downsampling by a substantial margin as the reduction ratio increases.

accuracy achieved by ToMe at a much lower 10% reduction.

Figure 4 (b) compares the throughput of the two methods. STORM achieves higher throughput across all reduction ratios, which results from its structured design. ToMe relies on global similarity computations over a flattened sequence, creating a scalability bottleneck. STORM decomposes reduction into parallel row/column operations, replacing a single large computation with many smaller independent ones. This design not only preserves spatial integrity but also reduces overall complexity, enabling both stronger robustness and faster inference. We provide corresponding results of EViT on Appendix 7.3.

*Table 3.* Performance comparison with Mamba-specific token reduction methods on ImageNet-1K classification under training-free settings. The "+" symbol indicates the integration of a token reduction method into VMamba. STORM achieves the highest top-1 accuracy across all model scales with comparable FLOPs.

| Method | Acc1 (%) | | GFlops | |
| --- | --- | --- | --- | --- |
| | small | base | small | base |
| VMamba (baseline) | 83.7 | 83.9 | 8.72 | 15.36 |
| +HSA | 46.2 | 45.0 | 6.10 | 11.00 |
| +MTR | 47.9 | 49.4 | 6.29 | 11.08 |
| +QuarterMap | 64.2 | 69.9 | 6.22 | 11.01 |
| +STORM (ToMe) | **82.0** | **83.3** | 6.11 | 11.00 |

This divergence stems from the fundamental difference in reduction strategy. Pooling and downsampling perform fixed, local aggregation regardless of content, which preserves spatial regularity but inevitably discards fine-grained information. In contrast, STORM retains content-aware selection while enforcing spatial organization. This allows it to maintain both structural integrity and semantic fidelity under aggressive compression, leading to superior performance where simple resolution scaling fails.

**Robustness under fine-tuning.** Table 5 presents the fine-tuning results of different token reduction methods on VMamba-T and VMamba-S under varying reduction ratios. STORM consistently achieves the highest accuracy across both backbones and all reduction ratios after fine-tuning, substantially outperforming the baseline ToMe. For instance, on VMamba-T at a 0.32 reduction ratio, ToMe recovers to only 32.7%, while STORM reaches 78.4%. A similar trend holds on VMamba-S, where STORM achieves 80.4% at a 0.36 reduction ratio compared to 31.3% for ToMe. These results demonstrate that STORM maintains strong robustness under fine-tuning, effectively recovering performance even under aggressive token reduction.

*Table 4.* Comparison of token reduction strategies under training-free settings. Random pruning results are averaged over five runs. STORM significantly outperforms conventional methods in both guided and random reduction, confirming its inherent robustness.

| Method | Reduction Ratio | GFlops | Acc1 (%) | Δ (%) |
| --- | --- | --- | --- | --- |
| | 0 | 8.72 | 83.7 | 0 |
| | 0.18 | 5.57 | 32.9 | 0 |
| ToMe | 0.23 | 4.87 | 23.3 | 0 |
| | 0.29 | 4.07 | 17.6 | 0 |
| | 0.36 | 3.40 | 11.7 | 0 |
| | 0 | 8.72 | 83.7 | 0 |
| | 0.18 | 5.31 | 79.0±0.3 | 46.1↑ |
| STORM (random) | 0.23 | 4.61 | 77.1±0.5 | 53.8↑ |
| | 0.29 | 3.81 | 74.4±0.9 | 56.8↑ |
| | 0.36 | 3.15 | 69.1±1.0 | 57.4↑ |
| | 0 | 8.72 | 83.7 | 0 |
| | 0.18 | 5.31 | **80.9** | **48.0↑** |
| STORM (ToMe) | 0.23 | 4.61 | **80.0** | **56.7↑** |
| | 0.29 | 3.81 | **78.0** | **60.4↑** |
| | 0.36 | 3.15 | **73.3** | **61.6↑** |

### 4.5. Ablation Study

**Effect of structured reduction.** Figure 6 compares the conventional ToMe method with STORM (ToMe)-w/o win, a structured variant that reduces tokens within rows and columns while omitting local constraints. The results demonstrate that the structured variant achieves dramatically higher accuracy than ToMe across all reduction ratios. For instance, at a 29% reduction, it retains 71.4% accuracy compared to only 17.6% for ToMe.

This decisive improvement demonstrates that the core effectiveness of our framework stems from its structured spatial reduction paradigm. By operating on organized rows and columns instead of a globally flattened sequence, the method inherently preserves the 2D token grid. This fundamental spatial alignment effectively prevents the performance col-

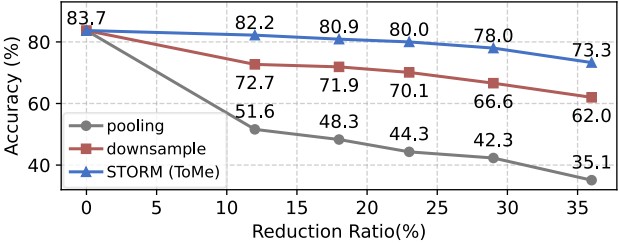

*Figure 5.* Comparison of STORM with conventional spatial reduction methods across varying token reduction ratios. By integrating structured reduction with content-aware token selection, STORM maintains significantly higher accuracy under aggressive compression, demonstrating superior robustness.

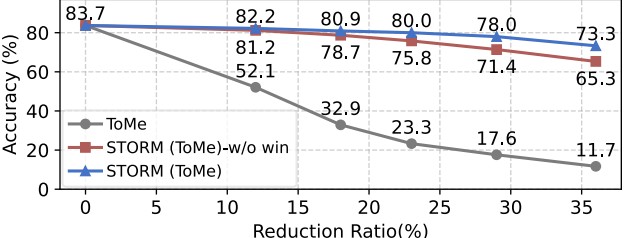

*Figure 6.* Ablation study on STORM. STORM (ToMe)-w/o win denotes the variant with structured spatial reduction only, without the localized windowing constraint. STORM significantly outperforms ToMe, and performance is further improved when the windowing constraint is applied.

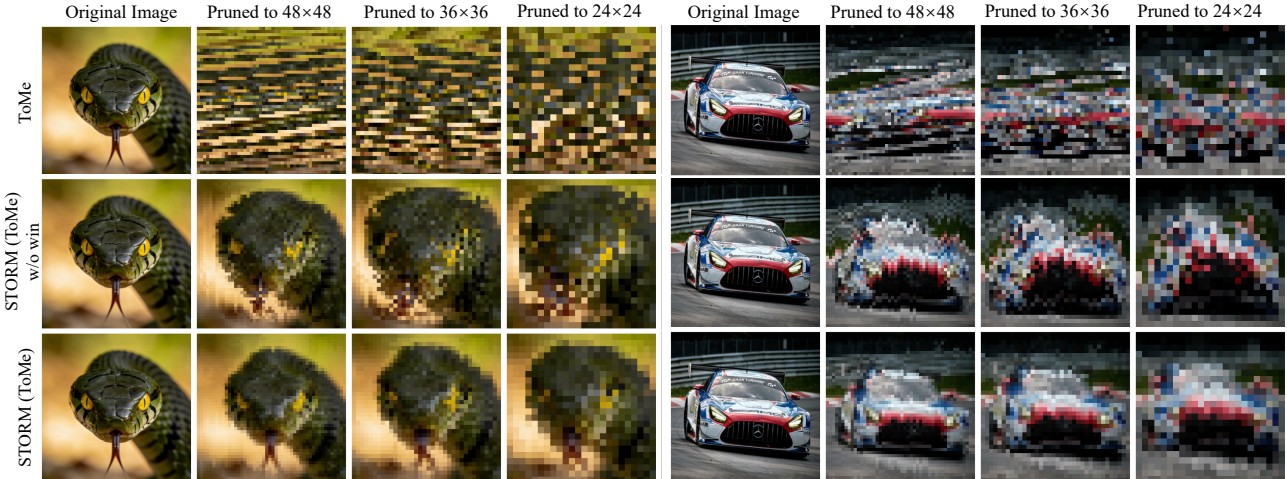

*Figure 7.* Visualization of reduction tokens at varying compression ratios. ToMe produces fragmented and spatially inconsistent representations. Structured spatial reduction (without windowing) restores layout regularity but sacrifices fine-grained details. In contrast, the full STORM framework consistently preserves both structural integrity and semantic coherence across all pruning levels.

*Table 5.* Comparison of token reduction methods after fine-tuning on VMamba-T and VMamba-S. All methods are fine-tuned for 5 epochs with a learning rate of 1e-5. STORM consistently outperforms ToMe across all reduction ratios, demonstrating strong recoverability after fine-tuning.

| Method | Reduction Ratio | GFlops | Fine-tuned Acc1 (%) |
|---|---|---|---|
| VMamba-T | 0 | 4.91 | – |
| +ToMe | 0.20 | 2.97 | 47.9 |
| +ToMe | 0.32 | 2.11 | 32.7 |
| +STORM (ToMe) | 0.20 | 2.71 | **80.7** |
| +STORM (ToMe) | 0.32 | 1.86 | **78.4** |
| VMamba-S | 0 | 8.72 | – |
| +ToMe | 0.23 | 4.87 | 48.1 |
| +ToMe | 0.36 | 3.40 | 31.3 |
| +STORM (ToMe) | 0.23 | 4.61 | **82.1** |
| +STORM (ToMe) | 0.36 | 3.15 | **80.4** |

lapse inherent in conventional reduction approaches.

**Effect of window constraints.** Figure 6 compares the complete STORM framework with its ablated version that removes local windowing (STORM (ToMe)-w/o win). The results show that our full method consistently achieves higher accuracy, with the performance gap widening under more aggressive reduction. For example, at a 36% reduction ratio, the complete framework retains 73.3% accuracy, while the variant without windows drops to 65.3%.

This demonstrates that localized constraints are vital under high compression. While structured reduction preserves global scanning paths, windowing maintains local semantic coherence by preventing cross-region merging. This combination of global structure and local fidelity is key to

robust performance during extreme pruning. We include the ablation study of window size in the Appendix 7.5.

### 4.6. Visualization

Figure 7 visualizes the token groups retained after compression. The results show that ToMe produces scattered and semantically inconsistent clusters, often breaking objects apart. The structured variant organizes tokens into a regular grid, restoring spatial order but blurring fine details. In contrast, STORM maintains semantically coherent groups where object outlines remain sharp and local features are preserved, demonstrating its effectiveness in balancing structural integrity with semantic fidelity.

## 5. Conclusion

This paper identifies that the failure of token reduction in vision Mamba stems from the disruption of 2D spatial structure. We propose STORM, a plug-and-play framework that enforces structured spatial reduction with local window constraints, preserving both global grid topology and local semantic coherence. Without training, STORM enables robust and aggressive compression and achieves superior performance across diverse architectures and tasks.

## Acknowledgment

This work was supported by National Major Scientific Instrument and Equipment Development Project of National Natural Science Foundation of China under Grant 62427820; in part by National Natural Science Foundation of China under Grant 62306198.

## Impact Statement

This paper presents work whose goal is to advance the field of Machine Learning. There are many potential societal consequences of our work, none which we feel must be specifically highlighted here.

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

# 6. Implementation Details

## 6.1. Reduction Ratios Configuration

In our experiments, VMamba and LocalMamba share an identical pruning configuration: we designate the even-indexed blocks of Stages 1–3 as pruning anchors for VMamba. By uniformly distributing the token reduction budget across these anchors, we control the final output resolution of Stage 3 to $\{9 \times 9, 8 \times 8, 7 \times 7, 6 \times 6, 5 \times 5\}$, which corresponds to overall reduction ratios of $[0.12, 0.18, 0.23, 0.29, 0.36]$, respectively. Stage 4 is excluded from the pruning process, as its high semantic density and low spatial resolution lead to diminishing returns in computational efficiency gains while risking the loss of critical features.

In our PlainMamba experiments, we designate layers $[4, 10, 16, 22]$ as pruning anchors. By uniformly distributing the token reduction budget across these anchors, we control the equivalent final output resolution to $\{9 \times 9, 6 \times 6, 4 \times 4, 3 \times 3\}$, which corresponds to overall reduction ratios of $[0.25, 0.39, 0.52, 0.55]$, respectively.

## 6.2. Adaptation of EViT to Mamba

We follow HSA (Zhan et al., 2024a) to adapt EViT (Liang et al., 2022) to the Mamba architecture. The SSM block produces a selectivity parameter that governs input-dependent gating. Token importance is derived by aggregating this selectivity across the four scan directions of VMamba and averaging over the feature dimension, yielding a scalar score per token. Tokens with lower scores are treated as less informative and are prioritized for removal.

## 6.3. Non-Divisible Window Handling

We partition the feature map into spatial windows via sequential row/column splitting. When the spatial dimension is not divisible by the window size, the remaining tokens form a smaller window without padding or truncation. These irregular windows are processed identically to regular ones, as our scoring and pruning operations are agnostic to window shape.

## 6.4. Reduction Budget Allocation

The overall keep ratio $\rho \in (0, 1]$ is distributed across pruning anchors and their constituent windows. The total tokens to remove, $R_{\text{total}} = \lceil (1 - \rho) \cdot N \rceil$, is evenly partitioned among $K$ anchors, with any remainder assigned incrementally to the first anchors. Within each anchor, the budget is further subdivided across windows following a row/column-wise paradigm with uniform spacing, ensuring balanced spatial coverage. For multi-stage backbones, the per-anchor budget is scaled proportionally to the token count at each stage, distributing reduction pressure equitably across different spatial resolutions.

# 7. More Experiments

## 7.1. Image Classification on LocalMamba

*Table 6.* Performance comparison on ImageNet-1K. The symbol "+" denotes the application of token reduction methods to off-the-shelf models, with training-free accuracy reported. The symbol $\triangle$ indicates the performance difference between the baseline and the reduced model. STORM consistently yields substantial improvements over traditional reduction methods on LocalMamba.

| Method | GFlops | Params | Acc1(%) | $\Delta$(%) |
|---|---|---|---|---|
| LocalMamba | | | | |
| LocalVMamba-S | 11.37 | 50M | 83.7 | 0.0 |
| +EViT | 6.70 | 50M | 19.3 | 64.4↓ |
| +ToMe | 6.96 | 50M | 28.7 | 55.0↓ |
| +STORM (EViT) | 6.70 | 50M | **78.5** | **5.2↓** |
| +STORM (ToMe) | 6.70 | 50M | **79.6** | **4.1↓** |

Table 6 compares token reduction methods on LocalMamba-S under training-free settings. The results show that conventional methods cause severe performance degradation, with EViT and ToMe dropping accuracy to 19.3% and 28.7%, respectively. In contrast, STORM enables a strong recovery: STORM (ToMe) retains 79.6% accuracy, limiting the loss to only 4.1%.

This outcome further validates STORM's generalizability to Mamba variants with strong local inductive biases. LocalMamba

explicitly emphasizes neighborhood interactions, which are disrupted by global flattening in conventional reduction. STORM's row-column reduction preserves both local window structure and global layout, thereby maintaining the model's intended local-to-glue reasoning. The consistent success across VMamba, PlainMamba, and LocalMamba underscores that structured spatial preservation is a universally effective principle for token reduction in vision Mamba families.

*Table 7.* Performance comparison on object detection and instance segmentation on tiny-scale models. The symbol "+" denotes the application of token reduction methods to off-the-shelf models under training-free settings. STORM achieves the best performance across both tasks, with particularly strong results on the VMamba backbone.

| Method | $AP^b$ | $AP^b_{50}$ | $AP^b_{75}$ | $AP^m$ | $AP^m_{50}$ | $AP^m_{75}$ |
|---|---|---|---|---|---|---|
| VMamba | | | | | | |
| VMamba-T | 47.3 | 69.3 | 52.0 | 42.7 | 66.4 | 45.9 |
| +EViT | 2.7 | 7.9 | 1.4 | 1.0 | 3.0 | 0.5 |
| +ToMe | 2.6 | 7.9 | 1.2 | 0.9 | 3.2 | 0.3 |
| +STORM (EViT) | **43.1** | **68.2** | **48.5** | **39.1** | **65.1** | **41.8** |
| +STORM (ToMe) | **42.7** | **68.1** | **47.7** | **38.7** | **64.7** | **41.2** |
| PlainMamba | | | | | | |
| PlainMamba-L1 | 44.1 | 64.8 | 47.9 | 39.1 | 61.6 | 41.9 |
| +EViT | 30.9 | 48.9 | 32.4 | 27.8 | 46.2 | 28.7 |
| +ToMe | 31.1 | 49.1 | 32.9 | 28.1 | 46.5 | 28.9 |
| +STORM (EViT) | **38.2** | **59.9** | **40.8** | **34.3** | **56.1** | **36.2** |
| +STORM (ToMe) | **39.8** | **61.3** | **42.9** | **35.4** | **57.6** | **37.6** |

## 7.2. Downstream Tasks with Tiny-scale Models

Table 7 evaluates token reduction performance on smaller model variants, VMamba-T and PlainMamba-L1. The results show that conventional reduction methods lead to severe performance degradation, especially on VMamba-T where EViT and ToMe drop $AP^b$ to 2.7 and 2.6 respectively. STORM again delivers a strong recovery, raising $AP^b$ to 43.1 with STORM (EViT) and 42.7 with STORM (ToMe). A similar trend is observed on PlainMamba-L1, where STORM improves $AP^b$ by about 8–9 points over the baseline reduction methods.

These results demonstrate that STORM's effectiveness generalizes to more compact model architectures. Even with fewer parameters, the spatially sensitive modules in VMamba-T remain vulnerable to structural disruption from conventional reduction. By preserving the 2D token layout through row-column processing, STORM maintains the spatial alignment required by these modules, thereby enabling robust compression across different model scales. This consistent performance gain confirms that structured spatial reduction is a scalable and architecture-agnostic principle for efficient Mamba models.

## 7.3. Robustness across Various Reduction Ratios with EViT

Figure 8 (a) compares the accuracy of EViT and STORM (EViT) on VMamba-S across increasing reduction ratios. STORM maintains significantly higher accuracy under aggressive compression. Notably, at a 40% reduction rate, STORM retains

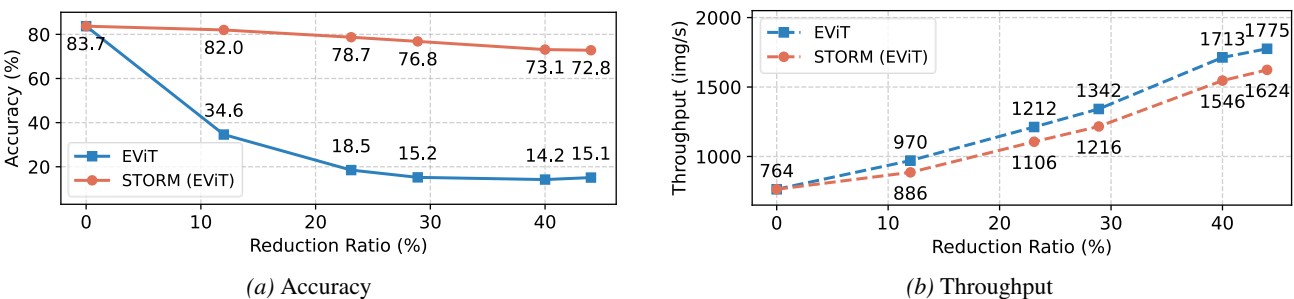

*(a)* Accuracy  *(b)* Throughput

*Figure 8.* Comparison with conventional token reduction methods on VMamba-S in terms of top-1 accuracy and throughput across varying reduction ratios. STORM achieves superior robustness with only a minor efficiency cost, delivering a better trade-off.

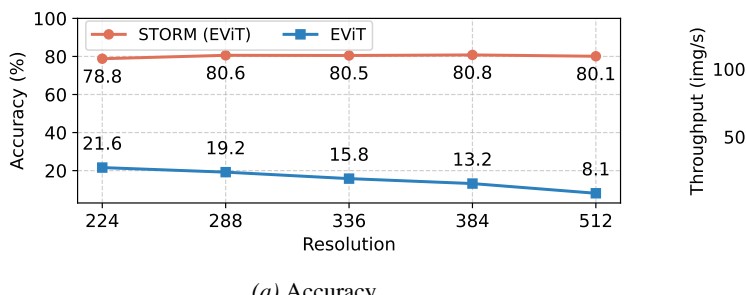 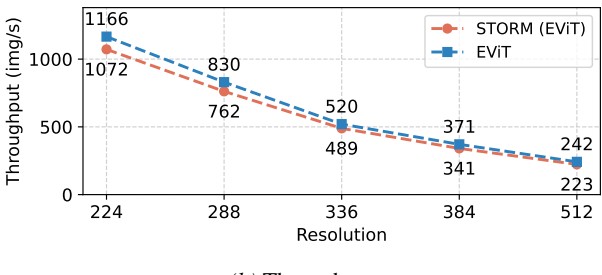

*(a)* Accuracy        *(b)* Throughput

*Figure 9.* Evaluation of STORM (EViT) on VMamba-S across different input resolutions. STORM achieves stable top-1 accuracy around 80% with only gradual throughput degradation, exhibiting superior robustness and a favorable trade-off compared to naive scaling.

72.8% accuracy, while EViT suffers a severe collapse to 15.2%.

Figure 8 (b) compares the throughput of the two methods. EViT achieves a modestly higher throughput than STORM across most ratios, which stems from the additional computational overhead of STORM's structured row-column operations. This minor efficiency cost, however, is justified by its dramatic accuracy advantage. By prioritizing the preservation of spatial integrity and model performance, STORM achieves the best accuracy-efficiency trade-off, delivering robust compression where conventional methods fail.

### 7.4. Robustness across Various Resolutions

Figure 9 (a) compares the top-1 accuracy of EViT and STORM (EViT) on VMamba-S across resolutions from 224 to 512. STORM maintains stable accuracy around 80.5%–81.1%, while EViT collapses from 21.6% at 224 to only 8.1% at 512. At 512 resolution, STORM outperforms EViT by nearly tenfold.

Figure 9 (b) compares throughput. EViT achieves higher throughput than STORM across all resolutions (e.g., 1166 vs. 1072 img/s at 224), but the gap gradually narrows as resolution increases (from 94 at 224 to 19 at 512). Despite STORM's modest efficiency disadvantage, its dramatic accuracy gain more than justifies the trade-off, delivering superior robustness where EViT fails.

### 7.5. Effect of Window Size

Window size is a key hyperparameter in STORM that balances local coherence with structured reduction. Figure 10 shows its impact on VMamba-S under two reduction ratios. As window size increases, accuracy consistently decreases for both settings, declining from 80.9% to 79.9% at ratio 0.18, and from 78.0% to 74.0% at ratio 0.29.

This trend confirms that localized windowing is most effective when token interactions are confined to compact neighborhoods. Small windows enforce strong semantic coherence among nearby tokens, preventing cross-region merging and preserving fine details. As windows expand, reduction decisions increasingly span distant regions, reintroducing interference and gradually converging toward the less robust behavior of the window-free variant. Therefore, maintaining a compact window is essential for STORM to achieve optimal robustness under aggressive compression.

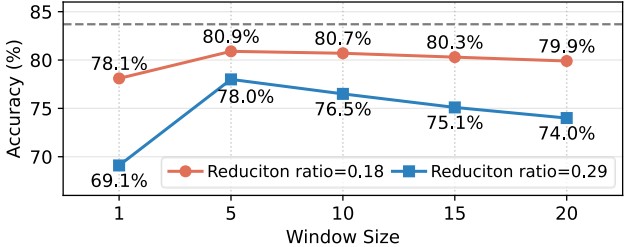

*Figure 10.* Effect of window size in STORM (ToMe) on VMamba-S. Accuracy consistently declines as the window expands under both reduction ratios, indicating that compact windows are essential for maintaining local semantic coherence during structured reduction.

## 8. Visualization

Figures 11 and 12 visualizes the token merging results of STORM (ToMe) under extreme compression, where the token layout is reduced from 26×26 to 6×6 (approximately 95% token reduction). The results show that patches from the same semantic region, such as object parts or continuous textures, are consistently merged into single tokens, forming spatially regular and semantically coherent groups.

This structural preservation under near complete token removal validates the core design of STORM. While conventional methods often fragment objects and disrupt spatial adjacency, STORM's row wise and column wise reduction maintains both neighborhood integrity and global layout regularity. The visualization therefore provides an intuitive explanation for STORM's robust quantitative performance, demonstrating that its structured approach retains essential visual semantics even under aggressive pruning.

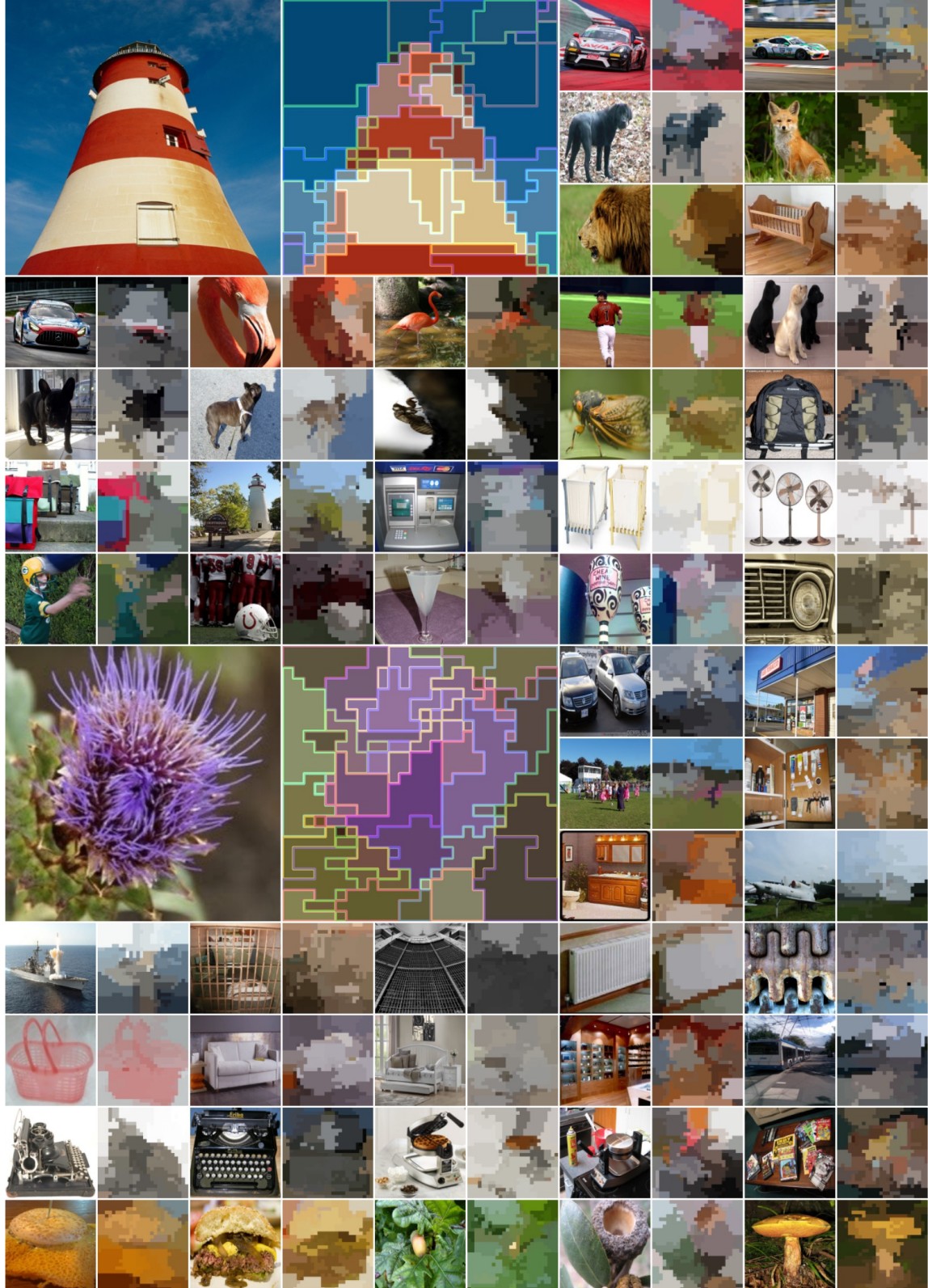

*Figure 11.* Visualization of extreme token reduction with STORM (ToMe). The figure illustrates the merging results on ImageNet-1k validation images when tokens are aggressively pruned from 26×26 to 6×6 (approximately 95% token reduction). Patches sharing the same color are merged into a single token, demonstrating how STORM preserves structural groups even under extreme compression.

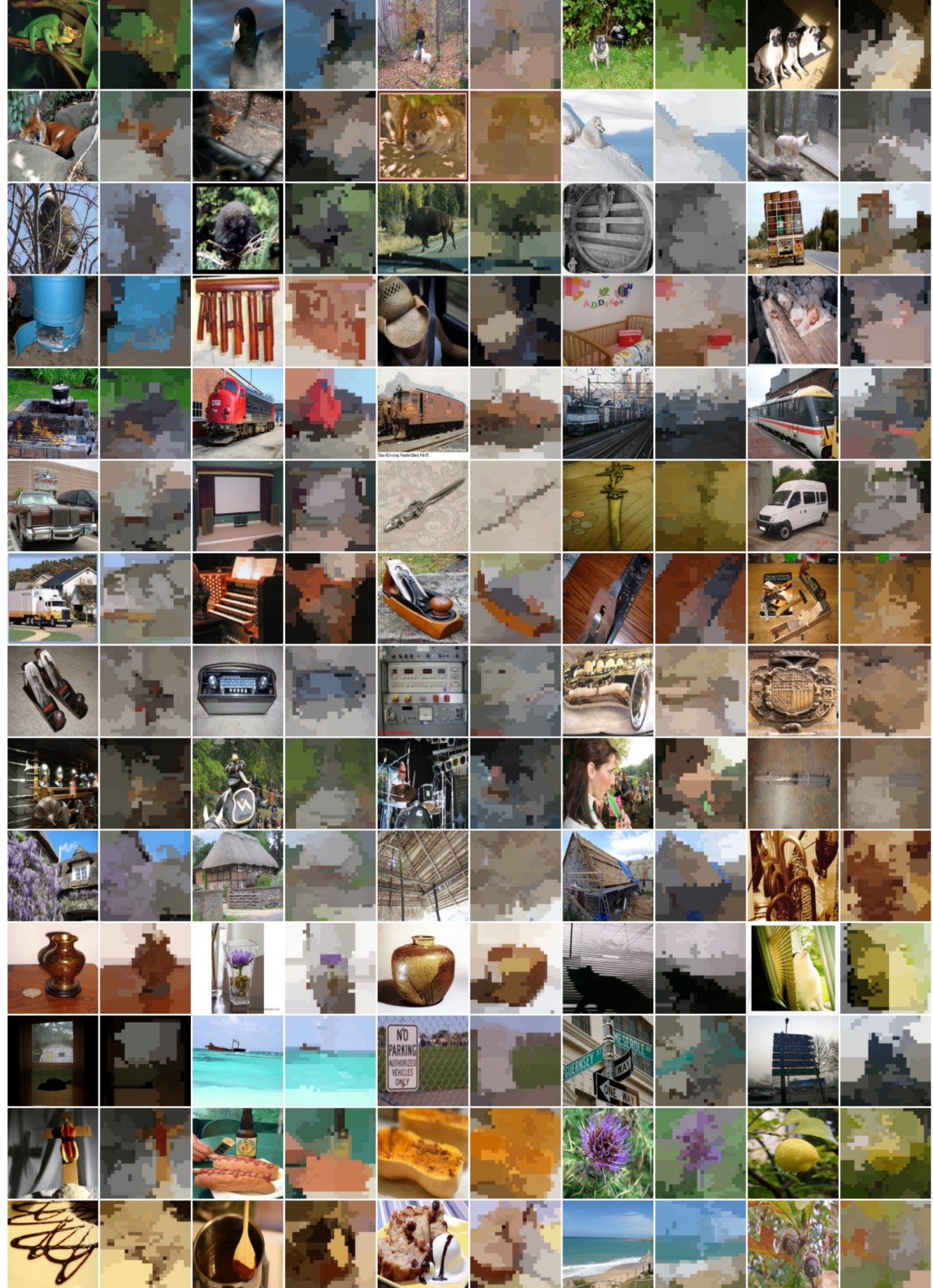

*Figure 12.* More visualization on images. Continuation of Figure 11

