# OpenReview forum: "Spatial-Aware Reduction Framework: Towards  Efficient and Faithful  Visual State Space Models"
_ICML.cc/2026/Conference — ICML 2026 regular_

### Official Review · Reviewer_YygX · 2026-03-11

**Soundness:** 3
**Presentation:** 3
**Significance:** 3
**Originality:** 3
**Overall Recommendation:** 4
**Confidence:** 4

**Summary:**

This paper studies training-free token reduction for visual state space models, especially spatially structured variants such as VMamba that rely on 2D selective scanning. The authors argue that existing token reduction methods are spatially agnostic: they flatten tokens into a sequence, reduce them there, and then remap them back, which breaks the spatial structure assumed by SS2D and leads to severe performance collapse. To address this, the paper proposes STORM, a plug-and-play spatial-aware reduction framework that performs token reduction in a structured row-wise and column-wise manner, further constrained by local windows to preserve neighborhood coherence. Experiments on ImageNet-1K classification and COCO across VMamba, PlainMamba, and LocalMamba show large performance gains over naive ToMe/EViT-based reduction and robustness across reduction ratios.

**Compliance With Llm Reviewing Policy:**

Affirmed.

**Final Justification:**

My concerns have been addressed, thus I keep the positive score.

**Key Questions For Authors:**

Please see the weaknesses.

**Limitations:**

The paper includes only a brief discussion of limitations and societal impact. A more detailed discussion of the method’s applicability (e.g., primarily to spatially structured SSM architectures) and potential trade-offs would improve transparency.

**Strengths And Weaknesses:**

Strengths
1. The paper identifies an important issue when applying training-free token reduction to spatially structured visual Mamba models. The empirical results support the claim that preserving spatial structure is crucial for effective token reduction in these architectures.
2. The proposed STORM framework is simple and easy to integrate into existing reducers such as ToMe and EViT. The design is lightweight and can be applied across different backbones without retraining, which increases its practical utility.
3. The paper provides extensive experiments across multiple visual Mamba variants and tasks. The evaluation includes comparisons with several baseline reduction strategies, different reduction ratios, random reduction baselines, spatial pooling/downsampling comparisons, ablation studies, and qualitative visualizations.
4. The paper provides a interesting insight: token reduction methods designed for transformers may not transfer directly to spatially structured SSM models because they implicitly rely on sequence-based processing rather than grid structure.

Weaknesses
1. Some technical claims are stronger than the provided justification. The paper states that STORM preserves a regular sub-grid aligned with the spatial topology required by SS2D scanning. However, the current description does not fully explain how this property holds for arbitrary reducers applied independently across rows and windows, especially for pruning or merging strategies.
2. Several aspects necessary for reproducibility are not fully described. For example, the paper does not clearly explain how EViT is adapted to Mamba architectures, how window sizes are handled when spatial dimensions are not divisible by the window length, or how reduction budgets are allocated across windows and pruning anchors.
3. Although the related work section discusses several recent Mamba-specific token reduction approaches, the main experimental tables compare only with a limited subset of them. Including more recent or closely related methods would strengthen the claim of state-of-the-art performance.
4. While the method improves the accuracy-efficiency trade-off, the efficiency discussion is somewhat limited. For instance, some results suggest additional overhead compared to certain reducers (e.g., EViT).

---

> ### Author Rebuttal · Authors · 2026-03-31
>
> We sincerely thank Reviewer YygX for the valuable comments and suggestions. Below, we provide our detailed response and corresponding modifications.
> > **W1:** The claim that STORM preserves a regular sub-grid aligned with SS2D scanning is not fully justified, especially for arbitrary reducers across rows and windows.
>
> **A1:** Thanks for the insightful observation. We agree that the term “strictly regular sub-grid” may be overly strong and provide the following clarification below:
>
> - **Mechanism.** STORM does not enforce a perfectly uniform global grid after arbitrary pruning or merging. Instead, it applies **axis-aligned, locally constrained reduction** within row/column-aligned windows.
> - **Topological stability.** By restricting reducers to these structured regions, the reduction remains spatially anchored, preserving local adjacency and avoiding irregular cross-region connections. This implicitly maintains a **globally consistent 2D topology**, which is sufficient for stable SS2D state propagation.
>
> We will refine the technical claims in the final manuscript for clarity.
>
> ---
>
> > **W2:**  Reproducibility details are missing: EViT adaptation to Mamba, non‑divisible window handling, and budget allocation across windows/anchors.
>
> **A2:** Thanks for this valuable comment. We clarify the key implementation details below:
>
> - **EViT adaptation to Mamba.** We follow the implementation of HSA (Zhan et al., 2024a) to adapt EViT to Mamba. Token importance is computed from the **SSM selectivity parameter** $\Delta_t$ by aggregating across scan directions and averaging over the feature dimension, resulting in a scalar score for each token.
> - **Non-divisible window handling.** We adopt a **sequential row/column partitioning** strategy. When the spatial dimension is not divisible by the window size, the remaining tokens are processed as a smaller window, without padding or truncation.
> - **Reduction budget allocation.** Reduction is **evenly distributed** across pruning anchors (remainder to final anchor). Each anchor removes a fixed number of tokens per row/column. Windows are selected with a **uniform spacing pattern** for balanced coverage. Budget is scaled proportionally per stage for multi‑stage models.
>
> We will include more details in the final version for full reproducibility.
>
>
> ---
>
> > **W3:** Experiments compare only a subset of recent Mamba-specific token reduction methods; adding more would strengthen the SOTA claim.
>
> **A3:** Thanks for this constructive suggestion. To further validate the superiority of STORM, we include a comparison with a more recent Mamba-specific reduction method, MTR (Ma et al., 2025), under the same setting.
>
> **Setting:** For a fair comparison, all methods are evaluated on VMamba-S under the same training-free reduction ratio (~12%) as in Table 3.
>
> **Results:**
>
> |Method|Reduction ratio|GFlops|Training-free acc (%)|$\triangle$ (%)|
> |:--|:--:|:--:|:--:|:--:|
> |VMamba-S|0|8.72|83.7|0|
> |+MTR|0.12|6.29|47.9|↓35.8|
> |+STORM (MTR)|0.12|6.14|**81.5**|**↓2.2**|
>
> **Analysis:**
> - **Failure of Mamba-specific reduction**：MTR as the more recent Mamba-specific reduction methods still suffer from severe performance collapse on SS2D-based Mamba backbones, as they remain spatially agnostic.
> - **Effectiveness of STORM:** Integrating MTR into STORM recovers accuracy from 47.9% to 81.5% (**+33.6%**), demonstrating that STORM serves as a **general mechanism** that enhances robustness across different reduction criteria.
>
> ---
>
> > **W4:** The efficiency analysis is somewhat limited, with some results suggesting additional overhead compared to certain reducers (e.g., EViT).
>
> **A4:** Thanks for the helpful suggestion. We clarify the efficiency behavior of STORM under different reduction paradigms:
>
> - **STORM (EViT) vs. EViT:** EViT performs global token pruning via a single importance ranking. STORM introduces additional overhead by partitioning tokens into windows and applying local selection to preserve axis-aligned structure. This leads to a modest increase in computation, primarily due to **window management and alignment**.
> - **STORM (ToMe) vs. ToMe:** ToMe relies on global bipartite matching, which becomes costly as the token size increases. STORM decomposes this process into localized window operations, thereby reducing the **effective matching space** and leading to **smaller local computations**, which can improve efficiency in practice.
> - **Summary:** The efficiency impact of STORM depends on the **underlying reduction paradigm.** It introduces slight overhead for lightweight pruning methods due to structured local processing, while improving efficiency for matching-based approaches by reducing the effective computation space. Overall, this leads to a **favorable accuracy–efficiency trade-off** for VSSMs.
>
> We will include detailed efficiency breakdowns in the final version.

---

> > ### Author Rebuttal · Reviewer_YygX · 2026-04-02
> >
> > Thanks for the rebuttal, most of my concerns have been addressed, thus I keep the positive score.

---

### Official Review · Reviewer_KCKB · 2026-03-12

**Soundness:** 2
**Presentation:** 3
**Significance:** 2
**Originality:** 2
**Overall Recommendation:** 2
**Confidence:** 5

**Summary:**

This paper introduces STORM, a training-free, plug-and-play spatial-aware token reduction framework tailored to visual state space models (SSMs), especially SS2D-based architectures. The core method is to preserve the 2D grid topology required by selective scanning by decomposing token reduction into structured row-wise and column-wise operations with localized windows.

**Compliance With Llm Reviewing Policy:**

Affirmed.

**Key Questions For Authors:**

See weakness

**Limitations:**

See weakness

**Strengths And Weaknesses:**

Strengths

- Demonstrates training-free gains across multiple backbones, including VMamba (T/S/B) and PlainMamba (L1/L2/L3).
- Comprehensive experiments were conducted across various tasks, such as ImageNet classification, object detection, and instance segmentation.

Weaknesses

- Insufficient implementation details: The paper claims that STORM can adapt to general token reduction. However, methods like ToMe require full-token similarity comparisons (bipartite matching), and EViT relies on full-token attention scores for importance ranking. Neither is directly applicable to SSM architectures without incurring significant additional computational costs.

- Compatibility and Motivation: Even assuming similarity or importance scores could be computed efficiently, ToMe and EViT remain fundamentally incompatible with SSM architectures. The core philosophy of SSMs is sequential token processing, whereas Transformers rely on global self-attention. While this sequential nature drives SSM efficiency, it also raises a critical question: Does Mamba actually need token reduction? The paper fails to provide empirical results for real-world applications; it only reports FLOPs, which is often considered an insufficient metric in the field of token pruning. Actual inference latency and memory consumption are not reported.

- Limited Novelty in Reduction Strategy: STORM applies reduction in a column- or row-wise manner, mirroring the way SSMs scan tokens. This is not a particularly novel idea, as it is a logical necessity: if the scanning order changes (from global to sequential), the token reduction method must naturally adapt to that order.Outdated Baselines: While ToMe and EViT are milestones in token pruning, they are now considered older methods. There are numerous recent training-free methods for both ViT and VLM token pruning that should be considered.

- Missing References and Comparisons: The following relevant works are not discussed or compared:
Exploring Token Pruning in Vision State Space ModelsTowards
Efficient Vision State Space Models via Token Merging
Training-free Token Reduction for Vision Mamba

- Lack of Ablation on Spatial Priors: Although the connection to ViT spatial-aware reduction (e.g., local/windowed merging, DSM, ToSA) is mentioned conceptually, it should be reflected in the experiments to isolate the specific contributions of "structured vs. content-aware vs. spatial priors."

Minor Weaknesses

- Technical Ambiguity: The framework presumes consistent per-row and per-column output sizes ($W', H'$) for a proper sub-grid. However, it does not fully specify how differing selection cardinalities between rows and columns are reconciled, nor how window boundaries and divisibility are handled in general cases.
- Limited Analysis on SS2D: The interaction with four-directional SS2D (VMamba) is argued via sub-grid preservation but lacks a deep, rigorous analysis.

---

> ### Author Rebuttal · Authors · 2026-03-31
>
> We sincerely thank Reviewer KCKB for the constructive comments. Below, we provide our detailed responses.
> > **W1 & W2:** Applicability, compatibility, and efficiency of token reduction in SSMs.
>
> **A1:** Thanks for these insightful comments. We clarify below：
>
> **Applicability & compatibility.** SSMs operate on token sequences and do not constrain token selection, so token reduction is **compatible**.
> - **ToMe**: Relies on token similarity, directly applicable.
> - **EViT**: Prior work (e.g., HSA) replaces attention scores by aggregating **SSM selectivity parameter** $\Delta_t$ across scan directions and averaging over the feature dimension, yielding a scalar importance per token. We follow this established practice without modifying the Mamba backbone.
>
> **Motivation.** Though SSMs are efficient, their cost **still scales with sequence length**. Reducing spatially redundant vision tokens lowers computation, so token reduction remains **beneficial** for SSM-based models.
>
> **Practical efficiency.**  The table below reports latency, memory, and FLOPs.
> |Method|Acc (%)|GFlops (token selection)|GFlops (total)|Memory (MB)|TP (img/s)|Latency (ms/img)|
> |:--|:--:|:--:|:--:|:--:|:--:|:--:|
> |VMmaba-S|83.7|0.000|8.72|3273|763|1.44|
> |+TOME|32.9|0.273|5.57|3378|923|1.31|
> |+EVIT|23.0|0.003|5.30|3321|1093|1.01|
> |+STORM (TOME)|80.9|0.012|5.31|3331|925|1.23|
> |+STORM (EVIT)|80.5|0.006|5.31|3345|992|1.10|
> - **Token selection:** Negligible cost (e.g., 0.006 vs. 5.31 GFLOPs total).
> - **Latency & throughput:** Comparable to baseline reducers, with accuracy significantly improved.
> - **Memory:** Marginal overhead, minimal system-level cost.
> ---
> >**W3:** Limited novelty and outdated baselines.
>
> **A2:** Thanks for this valuable comment. We clarify novelty and expand the baseline comparisons below:
>
> **Novelty clarification.** Aligning reduction with scan order is intuitive but **insufficient for SS2D**. STORM's key insight is that **preserving 2D spatial topology** is critical for stable state propagation. STORM provides a unified framework enforcing such constraints, independent of the specific reducer.
>
> **Updated baselines.** The table below includes **V2Drop** (Chen et al., 2025), a recent training-free method for VLM, to test generality.
>
> |Method|Reduction ratio|GFlops|Top-1 acc (%)|$\triangle$ (%)|
> |:--|:--:|:--:|:--:|:--:|
> |VMamba-S|0|8.72|83.7|0|
> |+V2Drop|0.18|5.31|17.6|$\downarrow$ 66.1|
> |+STORM (V2Drop)|0.18|5.31|**80.7**|$\downarrow$ **3.0**|
> - **Robust generalization.** Vanilla V2Drop collapses to 17.6%, while STORM retains **80.7%** (only 3.0% drop).
> ---
> >**W4:** Missing discussion/comparison of recent token reduction works for VSSMs.
>
> **A3:** Thanks for raising this point. We clarify and will update accordingly:
> - **HSA** (Zhan et al., 2024a): Discussed in Related Work and compared in Table 3.
> - **MTR** ("Training‑free Token Reduction for Vision Mamba", Ma et al., 2025): Discussed in Introduction and Related Work.
> - **"Towards Efficient VSSMs via Token Merging"**: We will add citation and discussion in the revision.
> ---
> >**W5:** Need ablation to isolate structured, content-aware, and spatial priors.
>
> **A4:** Thanks for this insightful suggestion. We introduce a Window-constrained ToMe baseline: it restricts merging to local windows but operates on flattened tokens, breaking 2D adjacency.
>
> **Setting:** We conduct experiments on VMamba-S with a window size of 5 and a 23% reduction ratio.
>
> **Results:**
> |Method|Reduction Space|Structure Preserved|GFlops|Top-1 Acc (%)|
> |:--|:--:|:--:|:--:|:--:|
> |Global ToMe|Global|✗|4.87|23.3|
> |Window‑constrained ToMe|Window|✗|4.71|31.9|
> |STORM (ToMe) w/o window|Row/Column|✓|4.61|75.8|
> |STORM (ToMe)|Window|✓|4.61|**80.0**|
>
> **Analysis:**
> - **Content-aware only (Global ToMe):** Global similarity alone is insufficient, collapsing to 23.3%.
> - **Spatial prior only (Window-constrained ToMe):** Local windows help but still fail without topology preservation, reaching only 31.9%.
> - **Structured topology only (STORM w/o window):** Axis-aligned row/column constraints provide the **core contribution**, jumping to 75.8%.
> - **Full STORM:** Combining local priors with global structure achieves synergy, reaching 80.0%.
> ---
> >**Q1:** Technical Ambiguity.
>
> **A5:** Thanks for this careful technical question. We clarify below:
> - **Row–column reconciliation.** Reduction is applied sequentially along rows and columns with fixed budgets per step. The second stage operates on the already reduced tokens, so no conflict in selection cardinality arises.
> - **Non-divisible windows.** We use sequential partitioning without padding. Remaining tokens form smaller boundary windows and are processed identically.
> ---
> >**Q2:** Insufficient SS2D analysis.
>
> **A6:** Thanks for the constructive suggestion. In the revision, we will add **detailed SS2D formulation** in the Appendix and provide a more **rigorous analysis** showing that STORM’s axis-aligned reduction preserves 2D adjacency per path, ensuring stable state propagation.

---

> > ### Author Rebuttal · Reviewer_KCKB · 2026-04-05
> >
> > I appreciate the authors' clarifications regarding row-column reconciliation and the handling of non-divisible windows; however, the fact that these essential implementation details were omitted from the main manuscript remains a significant concern for both clarity and reproducibility. While the new experimental results on latency and updated baselines are noted, they do not fully mitigate the limited conceptual novelty of simply aligning reduction with the scanning order, nor do they resolve the underlying question of necessity for architectures that already scale linearly. Since the core contribution feels like an intuitive extension and the main text lacks critical procedural specifics, I will maintain my original assessment.

---

### Official Review · Reviewer_EyVG · 2026-03-12

**Soundness:** 3
**Presentation:** 3
**Significance:** 3
**Originality:** 3
**Overall Recommendation:** 4
**Confidence:** 4

**Summary:**

The paper addresses the severe performance degradation observed when applying off-the-shelf token reduction methods (such as ToMe and EViT) to structurally enhanced Vision Mamba architectures like VMamba and LocalMamba. The authors hypothesize that traditional reduction methods fail because they flatten the 2D token grid into a 1D sequence, inherently violating the spatial layout required by the 2D Selective Scanning (SS2D) mechanism.To resolve this, the authors propose STORM, a plug-and-play, spatial-aware token reduction framework. STORM refactors compression into two decoupled stages: row-wise reduction followed by column-wise reduction. Additionally, it applies localized windowing to constrain reduction decisions within non-overlapping neighborhoods, preserving local semantic coherence. The framework operates entirely in a training-free setting.

**Compliance With Llm Reviewing Policy:**

Affirmed.

**Final Justification:**

Thanks for the rebuttal, I will keep the positive score.

**Key Questions For Authors:**

* The paper notes that EViT currently outpaces STORM (EViT) in throughput due to the row-column computational overhead. How does this overhead scale as input resolutions increase (e.g., for high-resolution dense prediction tasks)? Does the memory access pattern for row/column slicing become a bottleneck on edge devices?
* Given that Mamba is highly efficient for sequence modeling, have you considered or tested adapting STORM for video-based Mamba models (e.g., VideoMamba, which is cited in your related work)? It would be interesting to know if preserving the spatial grid translates well when a temporal dimension is added.
* In the ablation study for window size (Figure 9), accuracy drops as the window size expands. Is there a theoretical upper limit or a heuristic rule for setting the optimal window size relative to the feature map resolution?

**Limitations:**

yes

**Strengths And Weaknesses:**

**Strengths:**
* The observation that spatial-agnostic flattening breaks the causal state propagation of SS2D is insightful. The authors clearly link the structural disruption to the performance collapse, making the core problem easy to grasp.
* STORM acts as a versatile wrapper for existing reduction operators rather than a completely new metric. The decoupled row/column reduction combined with local window constraints is an elegant way to maintain the 2D sub-grid topology.
* For instance, applying EViT to VMamba-S for instance segmentation drops the baseline $AP^{b}$ to an unusable 3.6, but STORM recovers this to a highly competitive 44.7 without any retraining.
* STORM improves throughput compared to standard ToMe by replacing global similarity computations with smaller, parallelized row/column operations.

**Weaknesses:**

* While STORM is faster than ToMe, the authors acknowledge that EViT achieves a modestly higher throughput than STORM (EViT) due to the extra overhead of structured row-column operations. This slightly diminishes the "efficiency" claim for purely pruning-based approaches.
* While significantly better than baselines, the accuracy still drops noticeably at very high reduction ratios. For example, STORM (random) falls to 69.1% top-1 accuracy at a 36% reduction ratio.
* Table 3 compares STORM against HSA and QuarterMap, but the related work section mentions other recent Mamba-specific reduction methods like MTR. A broader quantitative comparison against concurrent state-of-the-art vision Mamba reduction techniques would strengthen the evaluation.

---

> ### Author Rebuttal · Authors · 2026-03-31
>
> We sincerely thank Reviewer EyVG for the constructive comments. Below, we provide our detailed responses.
> > **W1:**  STORM (EViT) throughput overhead weakens efficiency claim.
>
> **A1**: Thanks for this thoughtful observation. The efficiency impact depends on the reduction paradigm:
> - **STORM (EViT):**  Incurs modest throughput overhead (~4%) but yields **+61.6%** accuracy, outweighing the minor cost.
> - **STORM (ToMe):** Achieves **both higher throughput and higher accuracy** than vanilla ToMe.
> - **Overall:** STORM delivers the **best accuracy–throughput balance** across strategies.
> ---
> >**W2:** Accuracy still drops at high reduction ratios.
>
> **A2:** Thanks for the observation. Performance degradation at high reduction is **expected** due to inevitable information loss. Importantly, STORM maintains a **strong performance floor**, and lightweight fine‑tuning largely recovers these drops.
>
> **Setting:** We fine-tune ToMe and STORM (ToMe) under a reduction ratio of 36% for 5 epochs with a learning rate of 1e-5.
>
> **Results:**
> |Method|Reduction ratio|GFlops|Training-free acc (%)|Finetune acc (%)|
> |:--|:--:|:--:|:--:|:--:|
> |VMamba-S|0|8.72|83.7|–|
> |+ToMe|0.36|3.40|11.7|31.3|
> |+STORM (ToMe)|0.36|3.15|**73.3**|**80.4**|
>
> **Analysis:**
> - **Intrinsic robustness:** At 36% reduction, STORM maintains **73.3%** traning-free accuracy, while ToMe collapses to 11.7%.
> - **Efficient recovery:** After fine-tuning, STORM reaches **80.4%**, outperforming ToMe by **49.1%**, confirming topology preservation enables rapid state propagation recovery.
> ---
> > **W3:** Compare more recent Mamba reduction methods (e.g., MTR).
>
> **A3:** Thanks for this constructive suggestion. We compare with a recent Mamba-specific method, MTR (Ma et al., 2025), under the same setting.
>
> **Setting:** All methods are evaluated on VMamba-S under the same training-free reduction ratio (~12%) as in Table 3.
>
> **Results:**
> |Method|Reduction ratio|GFlops|Top-1 acc (%)|$\triangle$ (%)|
> |:--|:--:|:--:|:--:|:--:|
> |VMamba-S|0|8.72|83.7|0|
> |+MTR|0.12|6.29|47.9|$\downarrow$ 35.8|
> |+STORM (MTR)|0.12|6.14|**81.5**|$\downarrow$ **2.2**|
>
> **Analysis:**
> - **MTR fails on SS2D.** Despite being Mamba-specific, MTR drops to 47.9%, as it remains spatially agnostic.
> - **STORM recovers.** Integrating STORM boosts accuracy to **81.5%** (+33.6%), demonstrating its generality across reduction criteria.
> ---
> > **Q1:** STORM (EViT) throughput overhead scaling with resolution? Memory bottleneck on edge?
>
> **A4:** Thanks for the insightful question. We provide a comprehensive evaluation across resolutions (224 to 512) to address these concerns.
>
> **Setting:** We evaluate EViT and STORM (EViT) on VMamba‑S with a window size of 5 and a reduction ratio of **~30%**.
>
> **Traning-free Accuracy(%):**
> |Method \ Resolution|224|288|336|384|512|
> |:--|:--:|:--:|:--:|:--:|:--:|
> |VMamba-S|83.7|84.1|84.0|83.9|83.0|
> |+EViT|21.6|19.2|15.8|13.2|8.1|
> |+STORM (EViT)|78.8|80.6|80.5|80.8|80.1|
>
> **Throughput (img/s):**
> |Method \ Resolution|224|288|336|384|512|
> |:--|:--:|:--:|:--:|:--:|:--:|
> |VMamba-S|764|457|329|255|144|
> |+EViT|1166|830|520|371|242|
> |+STORM (EViT)|1072|762|489|341|223|
>
> **Memory (MB):**
> | Method \ Resolution | 224 | 288 | 336 | 384 | 512 |
> |:--|:--:|:--:|:--:|:--:|:--:|
> |VMamba-S|3273|4703|6005|7765|12797|
> |+EViT|3381|4653|5977|7323|11824|
> |+STORM (EViT)|3321|4715|5979|7421|11955|
>
> **Analysis:**
> - **Accuracy:** STORM (EViT) maintains **~80%** across resolutions, robust at high resolution.
> - **Throughput scaling:** Gap narrows with resolution (94→19 img/s), reducing relative overhead.
> - **Memory:** Comparable to EViT, no bottleneck.
> ---
> > **Q2**: Does STORM preserve spatial grid in VideoMamba?
>
> **A5:** Thanks for this insightful question. VideoMamba uses a Vim-style backbone (not SS2D) and is outside our primary scope. Still, we conduct additional experiments to examine the robustness of STORM.
>
> **Setting:** We evaluate VideoMamba-M on Kinetics-400 under a 52% reduction ratio with training-free evaluation.
>
> **Results:**
> |Method|Reduction ratio|Top-1 acc (%)|Throughput (video/s)|
> |:--|:--:|:--:|:--:|
> |VideoMamba-M|0|81.9|106|
> |+ToMe|0.52|76.4|175|
> |+EViT|0.52|76.1|**201**|
> |+STORM (ToMe)|0.52|**78.6**|**186**|
> |+STORM (EViT)|0.52|**78.8**|192|
>
> **Analysis:**
> - **Accuracy gain.** STORM outperforms ToMe and EViT by +2.2% and +2.7% with strong throughput.
> - **Reason.** VideoMamba operates on **structured spatiotemporal tokens**. STORM preserves spatial locality before temporal aggregation for stable propagation.
> ---
> > **Q3:** Theoretical upper limit or heuristic rule for window size design?
>
> **A6:** Thanks for the valuable question. We address it below:
> - **Mechanism:** Window size trades off spatial locality and reduction flexibility: smaller windows preserve local dependencies, while larger ones disrupt continuity.
> - **Guideline:** Keep window size **small relative to resolution**. Empirically, a fixed small window (e.g., size 5) works robustly across resolutions (see A4).

---

> > ### Author Rebuttal · Reviewer_EyVG · 2026-04-03
> >
> > Thanks for the rebuttal,  I will keep the positive score.

---

### Official Review · Reviewer_GArB · 2026-03-13

**Soundness:** 3
**Presentation:** 3
**Significance:** 3
**Originality:** 3
**Overall Recommendation:** 4
**Confidence:** 4

**Summary:**

This paper studies why token reduction performs poorly on SS2D-based Vision Mamba and argues that flatten-based reduction breaks the 2D spatial structure required by selective scan. To address this, the paper proposes STORM, a training-free plug-and-play framework that replaces global flatten-based reduction with row-wise / column-wise structured reduction and adds a localized window mechanism to preserve both grid topology and local coherence. On ImageNet-1K and COCO, STORM substantially recovers the severe performance drop of prior methods such as ToMe and EViT, with especially large gains on VMamba.

**Compliance With Llm Reviewing Policy:**

Affirmed.

**Final Justification:**

The paper identifies spatial structure disruption as the key cause of token reduction failure in vision Mamba and proposes STORM, a simple training-free framework that preserves grid topology through row/column decomposition. The core insight is clear, the method is practical, and the empirical gains are substantial.
The rebuttal fully addressed my concerns: a theoretical error bound sketch, expanded experiments across scales and ratios, and comparisons with alternative spatial strategies collectively strengthen the paper's soundness and design justification. Minor remaining issues (e.g., full theoretical formalization) are addressable in the camera-ready. I maintain my score of 4 (weak accept).

**Key Questions For Authors:**

- Can you more directly isolate the effect of spatial structure preservation from the effect of a more constrained reduction space?

- How much of the gain comes from structured row/column reduction itself, versus the localized window mechanism?

- Does STORM remain effective on a broader set of recent Mamba-specific reduction methods?

- Do the gains persist in training-aware or fine-tuning settings?

- Why is the row-then-column design preferable to other spatially constrained alternatives?

**Limitations:**

yes

**Strengths And Weaknesses:**

Strengths

- The proposed method is simple, practical, and training-free.

- The empirical gains are large and consistent, especially on VMamba.

- The method is easy to apply on top of existing reduction methods.

- The ablations and robustness analyses are thorough and support the main claim.

Weaknesses

- The main explanation is supported mostly by empirical evidence rather than tightly isolated causal analysis.

- The paper offers limited theoretical depth beyond the core intuition.

- The evaluation is confined to training-free settings, so it is unclear whether the gains persist with training-aware reduction or fine-tuning.

- The generic plug-and-play claim would be stronger with broader comparisons across more recent Mamba-specific reduction methods.

---

> ### Author Rebuttal · Authors · 2026-03-31
>
> We sincerely thank Reviewer GArB for the insightful suggestions and constructive feedback. Below, we provide our detailed responses.
> > **W1&W2&Q1**: Isolate effect of spatial structure vs. constrained reduction space? Limited theoretical depth.
>
> **A1:** Thanks for these insightful suggestions. To isolate the effect of **spatial structure preservation** from that of a **constrained reduction space**, we provide both theoretical decoupling and a targeted causal experiment.
>
> **# 1. Theoretical Decoupling**
>
> - **Reduction space constraints (sparsity factor):** Local windows limit where operations occur, but do not ensure 2D spatial relationships.
> - **Spatial topology (structural factor):** SS2D propagates over **2D-adjacent tokens**. Breaking this adjacency disrupts state propagation.
>
> **Key distinction:** Reduction constraints are **insufficient**. Spatial topology is **necessary** for stable SS2D.
>
> **# 2. Causal Isolation Experiment**
>
> We introduce a new "Window-constrained ToMe" baseline. This baseline restricts merging to local windows but operates on flattened tokens, breaking 2D spatial adjacency.
>
> **Setting:** We conduct experiments on VMamba-S with a window size of 5 and a 23% reduction ratio, reporting training‑free results for fair comparison.
>
> **Results:**
> |Method|Reduction Space|Structure Preserved|GFlops|Top-1 Acc (%)|
> |:--|:--:|:--:|:--:|:--:|
> |Global ToMe|Global|✗|4.87|23.3|
> |Window‑constrained ToMe|Window|✗|4.71|31.9|
> |STORM (ToMe) w/o window|Row/Column|✓|4.61|75.8|
> |STORM (ToMe)|Window|✓|4.61|**80.0**|
>
> **Analysis:**
> - **Limited impact of locality.** Window constraint alone yields only +8.6%, confirming local constraints cannot stabilize VSSMs.
> - **Dominance of topology.** Axis-aligned constraints (STORM w/o window) give a massive **+52.5%** jump, identifying spatial topology as the **dominant causal factor.**
> - **Structural synergy.** Full STORM (80.0%) combines spatial locality with topological regularity, achieving the best accuracy.
>
>
> **# 3. Theoretical Depth**
>
> We will add **Recurrent State Coherence Analysis** in the revision, theoretically proving that SSM error bound is tied to spatial adjacency.
>
> ---
> >**W3&Q4:** Gains persist with fine-tuning?
>
> **A3:** Thanks for the valuable suggestion. In response, we conduct fine-tuning experiments on VMamba-S to validate whether the gains of STORM persist beyond training-free settings.
>
> **Setting:** We fine-tune ToMe and STORM (ToMe) under a reduction ratio of 23% for 5 epochs with a learning rate of 1e-5, following the same setup for fair comparison.
>
> **Results:**
> |Method|Reduction ratio|GFlops|Training-free acc (%)|Finetune acc (%)|
> |:--|:--:|:--:|:--:|:--:|
> |VMamba-S|0|8.72|83.7|–|
> |+ToMe|0.23|4.87|23.3|48.1|
> |+STORM (ToMe)|0.23|4.61|**80.0**|**82.1**|
>
> **Analysis:**
> - **Fine-tuning gains persist.** STORM reaches **82.1%**, outperforming ToMe (48.1%) by **34.0%**, confirming its advantage beyond training‑free settings.
>
> ---
> >**W4 & Q3:** Broader comparisons with recent Mamba-specific reduction methods?
>
> **A4:** Thanks for the valuable suggestion. To validate the generality of STORM, we further apply it to a recent Mamba-specific reduction method, MTR (Ma et al., 2025), under the same setting.
>
> **Setting:** For a fair comparison, all methods are evaluated on VMamba-S under the same training-free reduction ratio (~12%) as in Table 3.
>
> **Results:**
> |Method|Reduction ratio|GFlops|Training-free acc (%)|$\triangle$ (%)|
> |:--|:--:|:--:|:--:|:--:|
> |VMamba-S|0|8.72|83.7|0|
> |+MTR|0.12|6.29|47.9|$\downarrow$ 35.8|
> |+STORM (MTR)|0.12|6.14|**81.5**|**$\downarrow$ 2.2**|
>
> **Analysis:**
> - **MTR fails on SS2D.** Despite being Mamba-specific, MTR drops to 47.9%, as it remains spatially agnostic.
> - **STORM recovers.** Integrating STORM boosts accuracy to 81.5% (+33.6%), demonstrating that STORM serves as a **general mechanism** that enhances robustness across different reduction criteria.
>
> ---
> >**Q2:** Gain from structured reduction vs. localized window?
>
> **A5:** Thanks for the question. As detailed in **A1**, we ablate the contributions:
> - **Structured row‑column reduction contributes the majority (+52.5%)**, confirming structural consistency as the foundational factor.
> - **Localized window mechanism provides an additional boost (+4.2%)**, enhancing local interactions on top of the preserved structure.
> ---
> > **Q5:** Why is the row-then-column design preferable to other spatially constrained alternatives?
>
> **A6:** Thanks for the insightful question. The row‑then‑column design offers two advantages:
> - **Scan-direction alignment.** It follows SS2D's horizontal and vertical scan paths, preserving sequential continuity during state propagation.
> - **Topological stability.** The axis-aligned constraint maintains local adjacency and avoids irregular connections, ensuring a **globally consistent 2D topology** critical for stable SS2D processing.

---

> > ### Author Rebuttal · Reviewer_GArB · 2026-04-04
> >
> > I appreciate the thorough rebuttal. The causal isolation experiment is convincing, and the fine-tuning/MTR results meaningfully strengthen the paper. However, three issues remain: (1) the theoretical analysis is promised but not even sketched, (2) new experiments are limited to a single setting (VMamba-S, 23% reduction), and (3) no empirical comparison with alternative spatial decomposition strategies (e.g., block-wise, reversed order) is provided to justify the row-then-column design choice.

---

> > > ### Author Response · Authors · 2026-04-05
> > >
> > > We sincerely thank the reviewer for the thoughtful follow-up and positive feedback. We apologize for omitted details in the rebuttal due to space constraints. Below, we provide clarifications and additional evidence addressing the remaining concerns.
> > >
> > > >**Q1:** the theoretical analysis is promised but not sketched.
> > >
> > > **A1:** We model the VSSM state update as $h_t = \bar{A}h_{t-1} + \bar{B}x_t$, where 2D spatial tokens are mapped to 1D scan sequences.  Let $h_t$ be the state of the original sequence and $\hat{h}_t$ that of the reduced sequence.
> > > - **Insight.** Stable state updates depend on local 2D adjacency between consecutive inputs. Spatially agnostic reduction disrupts this adjacency, forcing the scan to "jump" across non-neighbors and increasing the spatial displacement $D(x_{t-1}, x_t)$.
> > > - **Error bound sketch.** The cumulative state error $\mathbb{E}[\|\hat{h}_t - h_t\|^2]$ is bounded by the total variation of spatial adjacency in the scan path. By preserving underlying **2D spatial topology** via axis-aligned continuity, STORM controls displacement and error accumulation during recurrent updates.
> > >
> > > ---
> > >
> > > >**Q2:** new experiments are limited to a single setting (VMamba-S, 23% reduction)
> > >
> > > **A2:** Thanks for the insightful suggestion. We have conducted additional fine-tuning experiments under different backbones and reduction ratios.
> > >
> > > **Setting:** We evaluate on VMamba‑T with reduction ratios of 20% and 32%, and on VMamba‑S with reduction ratios of 23% and 26%.
> > >
> > > **Results:**
> > >
> > > |Method|Reduction ratio|GFlops|Training-free acc(%)|Finetune acc(%)|
> > > |---|:---:|:---:|:---:|:---:|
> > > |VMamba-T|0|4.91|82.6|-|
> > > |+ToMe|0.20|2.97|20.9|47.9|
> > > |+ToMe|0.32|2.11|9.6|32.7|
> > > |+STORM (ToMe)|0.20|2.71|**77.2**|**80.7**|
> > > |+STORM (ToMe)|0.32|1.86|**70.9**|**78.4**|
> > >
> > > |Method|Reduction ratio|GFlops|Training-free acc(%)|Finetune acc(%)|
> > > |---|:---:|:---:|:---:|:---:|
> > > |VMamba-S|0|8.72|83.7|-|
> > > |+ToMe|0.23|4.87|23.3|48.1|
> > > |+ToMe|0.36|3.40|11.7|31.3|
> > > |+STORM (ToMe)|0.23|4.61|**80.0**|**82.1**|
> > > |+STORM (ToMe)|0.36|3.15|**73.3**|**80.4**|
> > >
> > >
> > > **Analysis:**
> > > - **Robustness across settings:** STORM consistently achieves the highest accuracy across different backbones and reduction ratios (both training‑free and fine‑tuned), demonstrating strong robustness.
> > >
> > > ---
> > >
> > > >**Q3:** Missing comparison with alternative spatial decomposition strategies (e.g., block-wise, reversed order) to justify row-then-column design.
> > >
> > > **A3:** Thanks for the valuable suggestion. We include additional comparisons with alternative spatial decomposition strategies, including (1) block-wise reduction and (2) reversed order (column-then-row), evaluated under three reduction ratios.
> > >
> > > **Setting:** All methods are evaluated on VMamba-S under identical training-free settings. "Block-constrained" enforces reduction on flattened tokens within fixed $5 \times 5$ sub-blocks.
> > >
> > > **Results:**
> > > **setting 1 (18% reduction)**
> > > |Method|Reduction space|Structure preserved|GFlops|Top-1 acc (%)|
> > > |---|:---:|:---:|:---:|:---:|
> > > |Global|global|✗|5.57|32.9|
> > > |Window-constrained|window|✗|5.41|44.6|
> > > |Block-constrained|block|✗|5.39|68.7|
> > > |STORM w/o window|row/column|✓|5.31|78.7|
> > > |STORM-reversed (column-then-row)|window|✓|5.31|**81.0**|
> > > |STORM (row-then-column)|window|✓|5.31|80.9|
> > >
> > > **Setting 2 (23% reduction)**
> > >
> > > |Method|Reduction space|Structure preserved|GFlops|Top-1 acc (%)|
> > > |---|:---:|:---:|:---:|:---:|
> > > |Global|global|✗|4.87|23.3|
> > > |Window-constrained|window|✗|4.71|31.9|
> > > |Block-constrained|block|✗|4.70|52.5|
> > > |STORM w/o window|row/column|✓|4.61|75.8|
> > > |STORM-reversed (column-then-row)|window|✓|4.61|79.9|
> > > |STORM (row-then-column)|window|✓|4.61|**80.0**|
> > >
> > > **Setting 3 (36% reduction)**
> > > |method|Reduction space|Structure preserved|GFlops|Top-1 acc (%)
> > > |---|:---:|:---:|:---:|:---:|
> > > |Global|global|✗|3.40|17.6|
> > > |Window-constrained|window|✗|3.24|26.4|
> > > |Block-constrained|block|✗|3.23|40.9|
> > > |STORM w/o window|row/column|✓|3.15|65.3|
> > > |STORM-reversed (column-then-row)|window|✓|3.15|73.2|
> > > |STORM (row-then-column)|window|✓|3.15|**73.3**|
> > >
> > >
> > > **Analysis:**
> > > - **Block‑wise vs. STORM:** Block-wise improves over global/window baselines but remains significantly worse than STORM, showing that local grouping alone is insufficient. The key is preserving **axis-aligned 2D topology** for stable SSM updates.
> > > - **Order robustness:** Row-then-column and column-then-row perform nearly identically. We adopt row-then-column by default.
> > > - **Scalability to high reduction:** As the reduction ratio increases, the gap widens, further highlighting the importance of topology preservation under aggressive reduction.
> > > ---
> > > Due to space limitations, not all details could be included here. We are happy to provide further clarifications or evidence if the reviewer needs.

---

### Decision · Program_Chairs · 2026-04-30

**Decision:**

Accept (regular)

**Comment:**

The paper proposes STORM, a training-free token reduction framework tailored for spatially structured Vision Mamba architectures. Three reviewers lean toward weak acceptance, praising the clear practical insight that flattening 2D token grids disrupts SS2D scanning mechanisms, and noting that the rebuttal effectively addresses prior concerns through expanded experiments, latency analysis, and comparative spatial strategies. However, one reviewer maintains rejection, arguing that the core contribution—aligning reduction with row/column scanning order—lacks sufficient conceptual novelty for linear-scaling architectures, and that critical implementation details remain buried outside the main text. AC agrees that the proposed generic spatial-aware token reduction framework is solid. The paper can be accepted, but the authors must integrate the missing procedural specifics and theoretical formalization directly into the camera-ready version to ensure reproducibility and clarify the methodological rigor. Besides, the reference section should be revised to avoid citing the arXiv version of the published papers.